## nature
## human behaviour
## OPEN
# Putative rhythms in attentional switching can be explained by aperiodic temporal structure

Geoffrey Brookshire [1,2] ✉

**The neural and perceptual effects of attention were traditionally assumed to be sustained over time, but recent work suggests that covert attention rhythmically switches between objects at 3–8 Hz. Here I use simulations to demonstrate that the analysis approaches commonly used to test for rhythmic oscillations generate false positives in the presence of aperiodic temporal structure. I then propose two alternative analyses that are better able to discriminate between periodic and aperiodic structure in time series. Finally, I apply these alternative analyses to published datasets and find no evidence for behavioural rhythms in attentional switching after accounting for aperiodic temporal structure. The techniques presented here will help clarify the periodic and aperiodic dynamics of perception and of cognition more broadly.**

Our senses provide us with a vast amount of simultaneous information. Attention helps focus perceptual processing on the important parts of a scene, boosting both perceptual sensitivity and neural responses to a stimulus[1]. The effects of attention were traditionally assumed to be sustained over short timescales; as long as someone holds their attention on a stimulus, processing of that stimulus is boosted. However, the field is now converging on a dramatically different view: that covert attention rhythmically switches between objects at 3–8 Hz (refs. [2–5]). Here I show that ubiquitous analyses in this literature conflate periodic oscillations with aperiodic temporal structure, leading to drastically inflated rates of statistical false positives. I then present two alternative analyses that can distinguish between periodic and aperiodic temporal structure in behaviour while controlling the rate of false positives. These methods could be applied to test for rhythms in the presence of structured noise, for any brief time series.

A growing behavioural literature argues that the focus of attention moves rhythmically between stimuli several times per second[6–26]. The experiments in this literature use a variety of stimuli but are built on a shared core design. In these studies, participants monitor two peripheral stimuli for a faint target. After a short delay, a cuing stimulus flashes on the screen. This cue has been hypothesized to reset the phase of low-frequency neural oscillations and serves to draw attention to one of the two peripheral stimuli. After a second variable delay, a faint target flashes on one of the peripheral stimuli. By averaging target-detection accuracy at each cue-to-target delay, these studies create a time course of attention towards the cued and uncued locations. To identify rhythms in attentional switching, amplitude spectra are then computed for these behavioural time courses. Peaks in the spectra are interpreted as evidence that attention moves rhythmically around the perceptual scene.

Studies of rhythmic attentional switching have used a wide range of different stimuli and dependent variables. For example, some studies examine visual attention to different spatial locations[6,7,9], whereas others focus on feature-based attention[20], global–local processing[27] or auditory attention[14]. These studies have reported rhythms in detection accuracy[6,7], reaction times[8], binocular rivalry[17], cue validity effects[25] and sensitivity and criterion metrics from signal detection theory[14]. Most studies reset ongoing dynamics with a cue stimulus, but some rely on participant-initiated

actions[11]. Although prominent theories focus on rhythms around 4–8 Hz (refs. [3–5]), studies in this literature have reported behavioural rhythms as low as 2.5 Hz (ref. [15]) and as high as 20 Hz (ref. [27]). This large and diverse literature is widely interpreted as convergent evidence for robust rhythms in attentional switching.

Here I demonstrate that the findings in this literature can be accounted for by attentional switching that is entirely non-oscillatory. Using computational simulations, I demonstrate that the spectral analyses used in this literature are sensitive not only to periodic rhythms but also to aperiodic temporal structure. I present two alternative methods that discriminate between periodic and autocorrelated aperiodic structure, control the rate of false positives, and recover true oscillations in behaviour.

## Results

**Identifying oscillations by shuffling in time.** Does attention move rhythmically between different objects? A large number of studies have addressed this question by searching for oscillations in densely sampled behavioural time series. After this time series has been converted to the frequency domain, oscillations appear as peaks in the amplitude spectrum. To interpret this spectrum, any putative oscillations must be discriminated from the background noise. How can we test whether a peak in the spectrum is significantly greater than the background noise?

Studies in this literature test for statistically significant oscillations by performing a randomization procedure that relies on shuffling the data in time. By shuffling in time, this analysis creates a surrogate distribution without any temporal structure and searches for oscillations against this surrogate distribution. I illustrate the basic procedure using details from an early influential study, Landau and Fries[6] (Fig. 1a). First, accuracy is computed at each time point, yielding a densely sampled behavioural time series. The data are then linearly detrended, multiplied by a Hanning taper and zero-padded, before the amplitude spectrum is computed with a discrete Fourier transform (DFT). To test whether peaks in this spectrum are statistically significant, a randomization test is performed. The time stamps of the raw behavioural data are shuffled a large number of times, and then the spectra of these time-shuffled data are computed. This results in a surrogate distribution of randomized spectra. For each frequency, the P value is computed as the

[1]Centre for Human Brain Health, University of Birmingham, Birmingham, UK. [2]SPARK Neuro, New York, NY, USA. ✉e-mail: brookshire@uchicago.edu

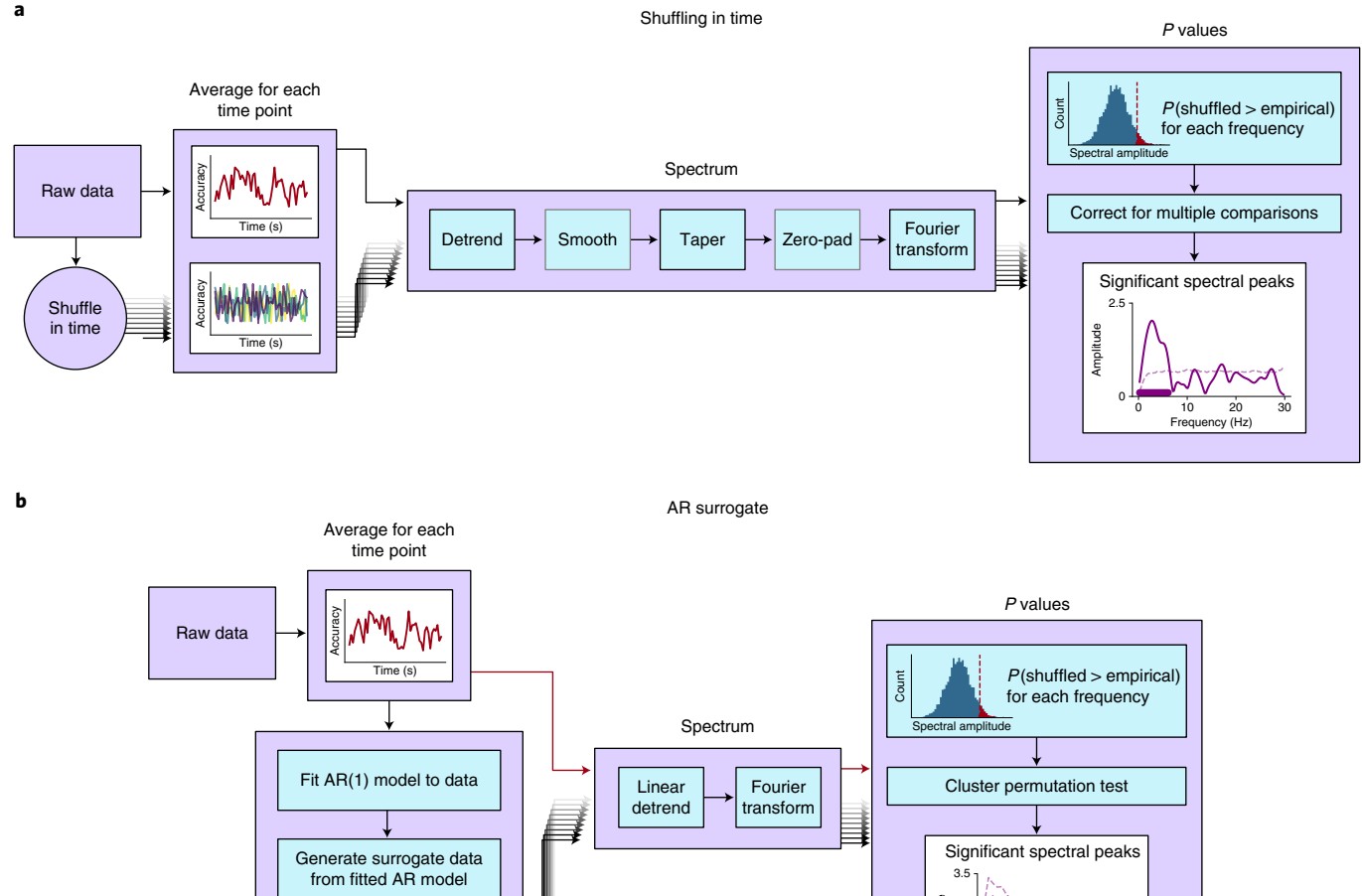

**Fig. 1 | Analysis pipelines for identifying oscillations in behaviour. a**, Standard analysis pipeline for identifying behavioural oscillations by shuffling in time. The analysis steps shown in grey are not performed in every study. Different researchers sometimes perform some steps in different ways—for example, by detrending with a second-order polynomial versus a sliding rectangular window. In the panel labelled 'Significant spectral peaks', the solid line indicates the empirical spectrum, the dashed line shows the significance threshold and the solid bar shows frequencies that were identified as statistically significant. **b**, AR surrogate analysis for identifying oscillations in behaviour.

proportion of randomized spectra with greater amplitude than the empirical spectrum. The *P* values are then corrected for multiple comparisons using Bonferroni corrections.

This shuffling-in-time procedure is widely used to study attentional switching[6–26] as well as other rhythms in perception[27–32]. The details of the analysis pipeline often differ between studies. For example, instead of computing accuracy at each time point, some studies use different dependent measures, such as average reaction time[8] or *d′* (ref. [14]). Some studies do not zero-pad the data before computing the amplitude spectrum[9], or they correct for multiple comparisons across frequencies using the false discovery rate (FDR)[7] or by selecting the largest peak in each shuffled spectrum[20]. All of these studies, however, determine statistical significance using a randomization test that shuffles the raw data in time.

**Shuffling in time alters aperiodic temporal structure.** Significant spectral peaks from the shuffling-in-time procedure are interpreted as reflecting periodic rhythms in attention. By shuffling the data in time, however, these studies test the null hypothesis that the behavioural data have no structure in time. These tests therefore do not provide unique evidence for oscillations in behaviour. Instead, they

provide evidence for any kind of structure in time. Randomization tests that shuffle the data in time reflect at least two varieties of non-oscillatory structure: aperiodic autocorrelation and consistency over trials.

First, shuffling in time destroys aperiodic temporal structure due to autocorrelation. Autocorrelation refers to correlations between a signal and lagged copies of itself. For periodic signals, the autocorrelation function has regularly spaced bumps, showing that the signal is positively correlated with itself at those periodic lags. For aperiodic signals, however, a different pattern emerges. For example, in a random walk, the signal at time *t* is strongly correlated with itself at time *t* − 1 but weakly correlated with itself at more distant times. As a consequence, the autocorrelation function smoothly drops down to zero with increasing lags. When data are simulated using a noisy random walk (an autoregressive model with a single positive coefficient (AR(1)); Fig. 2a), the autocorrelation function slowly drops to zero (Fig. 2c). After shuffling in time (Fig. 2b), however, the autocorrelation is approximately zero at all non-zero time lags (Fig. 2c). This difference in the autocorrelation functions also appears in the amplitude spectra. When those data are preprocessed using detrending and other common analysis steps, an

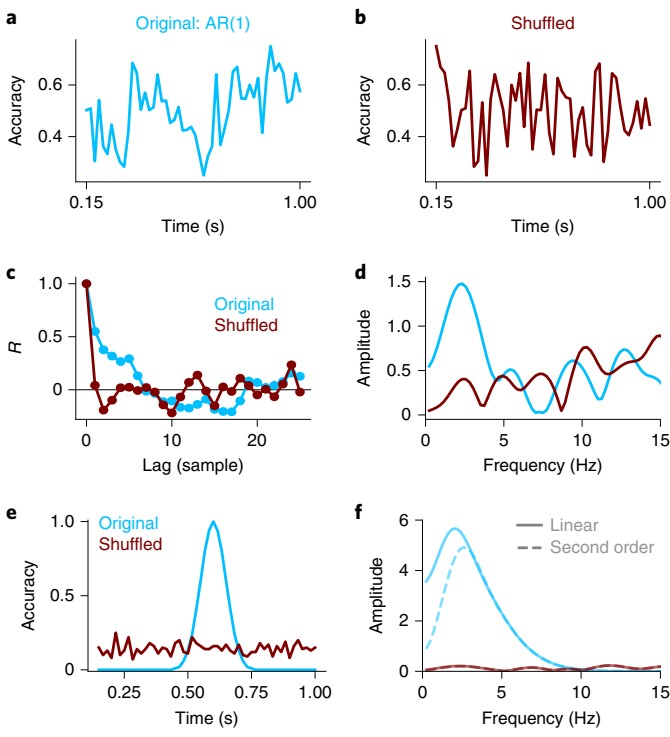

**Fig. 2 | Shuffling in time alters aperiodic temporal structure.**
**a–d**, Autocorrelation is destroyed by shuffling in time. Panel **a** shows simulated behavioural data generated with an aperiodic autoregressive model with a single positive coefficient (AR(1); $\beta = 0.5$). Panel **b** shows the same time series after being shuffled. Panel **c** shows the autocorrelation of the original AR(1) and shuffled time series. Panel **d** shows the amplitude spectra of the original AR(1) and shuffled time series, computed using the pipeline from Landau and Fries. Shuffling in time introduces an apparent peak in the spectrum of the AR(1) process, even though the autocorrelation function does not suggest that the signal is periodic. **e,f**, Consistency across trials is destroyed by shuffling in time. Panel **e** shows a simulated behavioural time series with a single peak in accuracy, and accuracy computed after shuffling trials. Panel **f** shows the amplitude spectra of the original and shuffled data, computed using the pipeline from Landau and Fries. The solid and dashed lines illustrate the results when preprocessed using linear and second-order polynomial detrending.

aperiodic autocorrelated signal often shows a single spectral peak, even though the data reflect a purely aperiodic process (Fig. 2d).

Second, shuffling in time destroys aperiodic consistency across trials. This is illustrated by simulating a behavioural time course with a single peak in accuracy (Fig. 2e). After the time points are shuffled between trials, that single peak in accuracy is spread out over all time points. In the amplitude spectra, this results in decreased amplitude at a wide range of frequencies, with a single peak that depends on the shape of the original aperiodic signal, as well as the preprocessing (for example, linear versus second-order polynomial detrending; Fig. 2f).

In summary, shuffling in time tests the null hypothesis that a time series contains no temporal structure whatsoever. However, this method cannot distinguish between periodic rhythms and aperiodic temporal structure.

**Distinguishing between periodic and aperiodic structure.**
Shuffling in time tests the null hypothesis that a time series has no temporal structure of any kind. Here I outline two procedures that can discriminate between periodic structure and autocorrelated aperiodic temporal structure in behavioural time series. The first procedure is a parametric bootstrap method based on

autoregressive models. The second procedure, based on harmonic analysis with the multi-taper method, is commonly used to identify rhythms in climate science[33].

The first method creates a surrogate distribution using an autoregressive model instead of by shuffling in time. I call this the 'AR surrogate' analysis (Fig. 1b). First, the empirical time series is obtained by computing accuracy (or some other aggregated measure) at each time point. Next, an autoregressive model with one positive coefficient (AR(1)) is fit to this time series. This AR(1) model captures the lag-1 autocorrelated aperiodic structure—but not the periodic structure—of the time series[33]. The fitted AR(1) model is then used to generate a large distribution of surrogate time courses. By comparing the empirical data with this surrogate distribution, we can test for oscillations against the null hypothesis that the data are lag-1 autocorrelated (but not periodic). For the empirical and surrogate time courses, the amplitude spectrum is obtained using a DFT after linearly detrending the data. $P$ values are computed for each frequency as the proportion of surrogate spectra with greater amplitude than the empirical spectrum (selecting non-DC frequencies below 15 Hz). Finally, a cluster-based permutation test[34] is used to correct for multiple comparisons across frequencies (though other corrections for multiple comparisons could also be used here).

For the second analysis method, I use a procedure that is widespread in climate science[33]. This method, called the 'robust estimate', was developed to identify oscillations in autocorrelated background noise in geological time series. Because human behaviour is also autocorrelated[35,36], this method may help distinguish behavioural rhythms from aperiodic background activity. This method uses multi-taper spectral analysis to compute the power spectrum of the signal, removes narrow-band peaks with median smoothing and then makes a robust estimate of the background noise by fitting an analytic AR(1) noise spectrum. Finally, statistical significance is computed by comparing the empirical spectrum to the AR(1) background fit[33].

**Shuffling in time inflates the rate of false positives.** Aperiodic temporal structure may appear as a peak in the amplitude spectrum when analysed by shuffling in time. To test how the different analysis methods reflect periodic and aperiodic structure, I simulated behavioural experiments of attentional switching. The experiments were simulated following the methods and analyses of two foundational studies in this literature: Landau and Fries[6], and Fiebelkorn et al.[7]. To explore the AR surrogate and robust estimate analyses, I simulated experiments following the behavioural paradigm in Landau and Fries.

To examine how aperiodic structure can lead to false positive results, I simulated four types of temporal structure. First, I simulated experiments in which every trial was independently and randomly determined to be a hit or a miss ('fully random'). These experiments had no temporal structure at all and act as a baseline for each analysis method. All four analysis methods yield false positives around or below the expected rate of $\alpha = 0.05$ (Fig. 3a and Table 1), indicating that they avoid false positives when testing against the null hypothesis that behaviour does not contain any temporal structure (but note that the false positive rate for Landau and Fries is slightly greater than 0.05).

To investigate how these methods perform with behaviour that is consistent across trials, I simulated experiments in which the response in each trial was randomly determined according to an idealized accuracy function. This function specified the time course of accuracy that would be obtained with an infinite number of trials. Experiments were simulated for three types of idealized accuracy time courses. In 'white noise' simulations, idealized accuracy time courses were generated with a random Gaussian process; these simulations included consistency over time but no other temporal structure. In 'random walk' simulations, idealized accuracy time

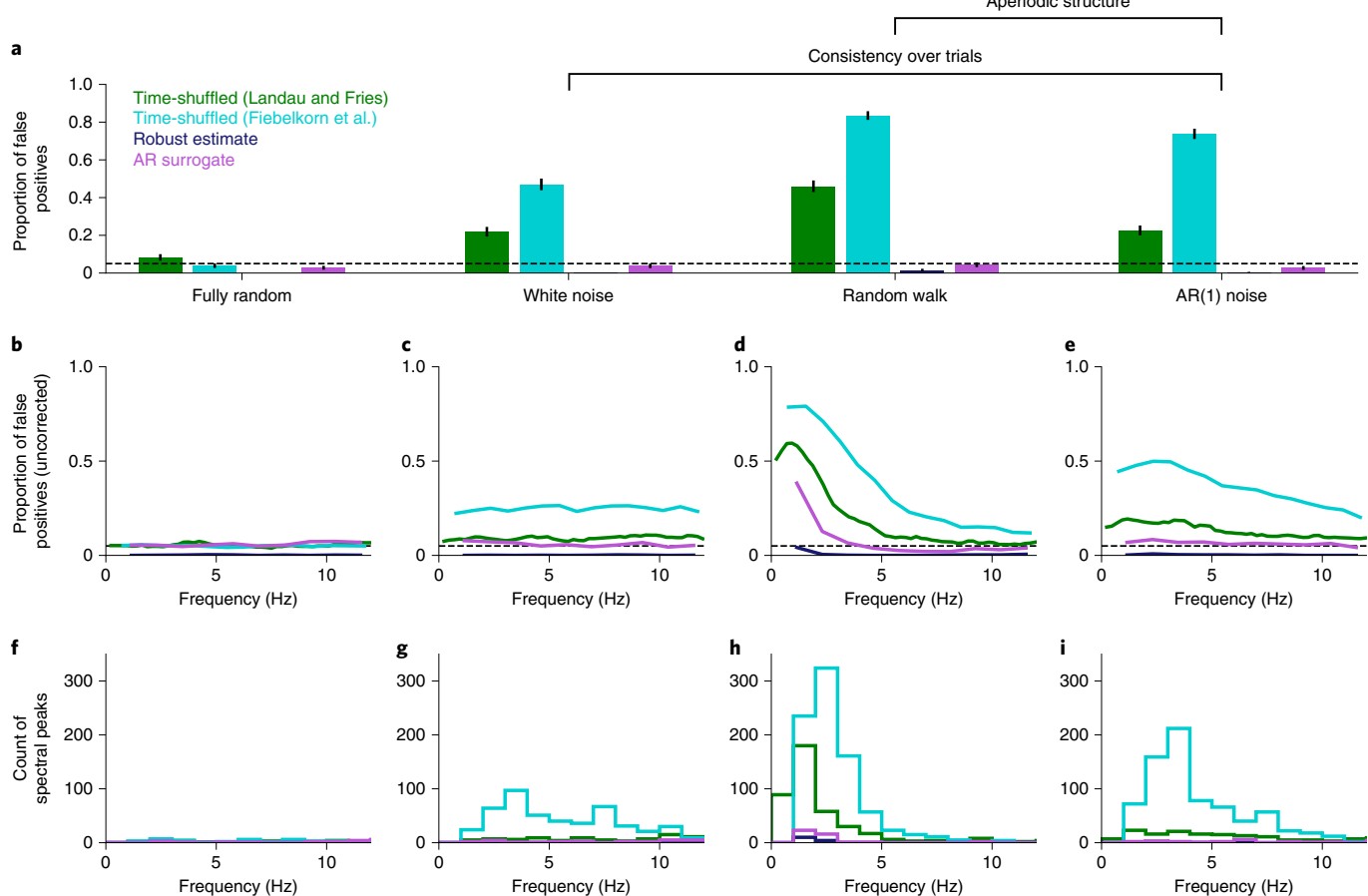

**Fig. 3 | False positive oscillations from aperiodic signals. a**, Proportion of false positive results in simulated experiments. All noise processes except fully random noise include consistency over trials, and non-zero autocorrelations are present in both random walk and AR(1) noise. The error bars show 95% confidence intervals. Each bar includes data from 1,000 simulated experiments. The dashed line shows the expected rate of false positives ($\alpha = 0.05$). **b–e**, The rate of false positives for each frequency bin, uncorrected for multiple comparisons. The dashed line depicts chance level (uncorrected, $\alpha = 0.05$). The exact frequency bins differ across methods due to differences in how each method computes the amplitude spectrum. **f–i**, Histograms of the spectral peaks in simulations ($k = 1,000$) with a statistically significant false positive result. The plots in **b–i** separately show results from each noise process: fully random (**b,f**), white noise (**c,g**), random walk (**d,h**) and AR(1) noise (**e,i**). The colours are the same as in **a**.

courses were generated with a random walk, including both consistency over trials and aperiodic temporal structure. Random walks, also called Brownian noise and $1/f^2$ noise, form the basis for various models in neuroscience[37] and psychology, including drift–diffusion models of decision-making[38]. Finally, in 'AR(1) noise' simulations, idealized accuracy time courses were generated with an autoregressive process with one coefficient ($\beta = 0.5$); these simulations include both consistency over trials and aperiodic temporal structure. These aperiodic noise functions are not intended to accurately model the dynamics of behaviour; instead, the goal is to test whether the different analysis methods can distinguish between oscillations and each type of aperiodic noise.

Analyses based on shuffling in time (Landau and Fries, and Fiebelkorn et al.) showed strongly inflated rates of false positives when the simulated data were consistent over trials (Fig. 3a, 'white noise'). False positive rates were higher still when the simulated data included autocorrelational structure (Fig. 3a, 'random walk' and 'AR(1) noise'). In contrast, the analyses that do not rely on shuffling in time (AR surrogate and robust estimate) had low rates of false positives for all noise types.

False positives when shuffling in time appeared at a wide range of frequencies, with the most common frequencies depending on the process used to generate the noise. To visualize these frequency profiles, we can plot the rate of significant results without correcting

for multiple comparisons over frequencies (Fig. 3b–e). In the white noise simulations, false positives were evenly distributed across frequencies (Fig. 3c), reflecting the flat spectrum of white noise. In the random walk simulations, false positives were biased towards lower frequencies, with a dip at very low frequencies (Fig. 3d) due to detrending before computing the Fourier transform. False positives in the AR(1) noise simulations were intermediate between the prior two noise types, with slightly higher rates of false positives at lower frequencies (Fig. 3e).

Do false positives appear at frequencies that are consistent with the reported literature? To answer this question, we can select experiments that show a statistically significant result after correcting for multiple comparisons over frequencies. We then plot the distribution of spectral peaks in those experiments (Fig. 3f–i). In the random walk simulations, false positive peaks tended to appear below 5 Hz but could also appear at higher frequencies (Fig. 3h). In the white noise and AR(1) noise simulations, false positive peaks appeared at a range of frequencies, including many within the theta band (Fig. 3g,i).

In summary, shuffling in time cannot distinguish between periodic and aperiodic temporal structure. Time-shuffled analyses can therefore lead us to conclude that behaviour is rhythmic even when that behaviour is generated using a purely aperiodic process. Brief aperiodic sequences often have amplitude spectra that appear plausibly rhythmic by eye; shuffling in time causes us to misidentify

**Table 1 | Proportion of false positives for each noise type and analysis method**

| Noise type | Analysis method | False positive rate | 95% confidence interval | P |
|---|---|---|---|---|
| Fully random | Landau and Fries | 0.083 | 0.067, 0.102 | $7 \times 10^{-6}$ |
| | Fiebelkorn et al. | 0.039 | 0.028, 0.053 | $\approx 1$ |
| | Robust estimate | 0.000 | 0.000, 0.004 | $\approx 1$ |
| | AR surrogate | 0.028 | 0.019, 0.040 | $\approx 1$ |
| White noise | Landau and Fries | 0.219 | 0.194, 0.246 | $3 \times 10^{-76}$ |
| | Fiebelkorn et al. | 0.470 | 0.439, 0.501 | $\approx 0$ |
| | Robust estimate | 0.001 | 0.000, 0.006 | $\approx 1$ |
| | AR surrogate | 0.038 | 0.027, 0.052 | $\approx 1$ |
| Random walk | Landau and Fries | 0.460 | 0.429, 0.491 | $4 \times 10^{-313}$ |
| | Fiebelkorn et al. | 0.835 | 0.811, 0.857 | $\approx 0$ |
| | Robust estimate | 0.015 | 0.008, 0.025 | $\approx 1$ |
| | AR surrogate | 0.044 | 0.032, 0.059 | 0.8 |
| AR(1) noise | Landau and Fries | 0.226 | 0.200, 0.253 | $2 \times 10^{-81}$ |
| | Fiebelkorn et al. | 0.738 | 0.710, 0.765 | $\approx 0$ |
| | Robust estimate | 0.003 | 0.001, 0.009 | $\approx 1$ |
| | AR surrogate | 0.027 | 0.018, 0.039 | $\approx 1$ |

The P values reflect the difference from the expected false positive rate of 0.05 (one-tailed binomial tests). $N = 1,000$ simulated experiments in each cell.

these spurious peaks as significant behavioural oscillations (Supplementary Fig. 1). In contrast, the two alternative analysis methods (AR surrogate and robust estimate) control the rate of false positives for aperiodic processes.

The robust estimate analysis is conservative across all types of aperiodic noise, showing a rate of false positives that is consistently below 0.05, and the AR surrogate method is conservative for some types of aperiodic noise (Fig. 3a). However, these methods must be preferred over shuffling in time, which inflates the rate of type I errors.

**Alternative methods recover true oscillations.** We can avoid false positive results by using analysis methods that do not rely on shuffling in time (the AR surrogate and robust estimate methods). Can these alternative methods also recover true oscillations in behaviour? To find out, I simulated experiments by combining random walk noise with sinusoidal modulation. These simulations examine the rate of statistically significant results as a function of the frequency and amplitude of behavioural oscillations. Amplitude was coded as the difference in accuracy between peaks and troughs of the idealized oscillation. For example, with amplitude 0.2, accuracy oscillates between 0.4 and 0.6.

These analyses report the proportion of simulated experiments that successfully recovered the behavioural oscillation. This measure is akin to an estimate of experimental power, assuming behavioural data include random walk background noise. Both the AR surrogate method and the robust estimate method successfully recover true oscillations in simulated behaviour. These methods most effectively recover oscillations at higher frequencies and higher amplitudes (Fig. 4a,b) and recover few behavioural rhythms below 4 Hz or below an amplitude of 0.2. The AR surrogate method outperformed the robust estimate method above 3 Hz, but the robust

estimate method performed slightly better at 2–3 Hz (Fig. 4c). Neither method reliably identified oscillations at 2–3 Hz, near the frequency resolution in these experiments.

How accurately does each analysis method reconstruct the frequency of oscillations in behaviour? For each experiment with a statistically significant result, we can compare the frequency of the true oscillation with the peak frequency in the amplitude spectrum. All four analysis methods were highly accurate for behavioural oscillations with amplitudes greater than 0.2 (Fig. 4d–g,l). Averaging over all frequencies and amplitudes, every analysis method had an average error of less than 0.3 Hz (Fig. 4m). This error is smaller than the frequency resolution of the AR surrogate and robust estimate analyses (1.15 Hz).

How do these alternative methods compare to the standard approach of shuffling in time? Shuffling in time could identify a large proportion of true oscillations, but results from these methods would be difficult to interpret if they also produced a large number of false positives. To quantify this trade-off, I computed the ratio of correct positive results to false positives when no oscillation is present ('detection ratio'). When the data were analysed by shuffling in time, the detection ratio was low (<3.5) for all frequencies and amplitudes (Fig. 4h,i). The AR surrogate and robust estimate analyses, however, showed much stronger detection ratios, especially at high frequencies and amplitudes (Fig. 4j,k). Aggregating over frequencies and amplitudes of simulated oscillations, both of the alternative methods showed substantially higher detection ratios than the methods that use shuffling in time (Fig. 4n and Table 2). The robust estimate method has the highest detection ratio, due to its exceptionally low rate of false positives (Fig. 3a and Table 1).

**False positives in published studies.** In principle, shuffling in time could cause researchers to find spurious rhythms in non-rhythmic behaviour. Alternatively, positive findings in this literature could reflect true attentional rhythms. To distinguish between rhythmic and aperiodic structure in prior studies, I reanalysed behavioural time courses in publicly available data[14,17,21,25]. These four published studies reported 11 statistically significant behavioural oscillations out of 23 tested time courses. When reanalysed using the AR surrogate and robust estimate methods, none of these tests reached statistical significance with either analysis method (Fig. 5). Although no statistical analysis can conclusively prove the absence of oscillations, the present results suggest that putative rhythms in behaviour could be explained by aperiodic temporal structure.

**Exploring the alternative methods.** How do the different methods behave with different experimental designs and analysis choices? The following simulations determine the proportion of false positive results under random walk noise, as well as the proportion of true positive results when the data are generated with an oscillation (frequency 6 Hz, amplitude 0.4, plus random walk noise). Because the rate of true positives depends on the frequency and amplitude of the behavioural oscillation (Fig. 4), these simulations should not be interpreted as an estimate of overall experimental power. Instead, they can help us understand how the analysis and experimental design influence the sensitivity and rate of false positives.

When the analysis only considers frequencies below 15 Hz (as in the analyses above), the AR surrogate method controls the rate of false positives (Table 1) and has a high proportion of true positive results (Fig. 6a). With a higher cut-off frequency, the AR surrogate method becomes more conservative (a smaller number of false positives; 15 versus 30 Hz: $\chi^2(1) = 8.5$; $P = 0.003$; Cramér's V ($\phi_C$) and 95% confidence interval, 0.07 (0.03, 0.11)) and less sensitive to true oscillations (15 versus 30 Hz: $\chi^2(1) = 258.1$, $P = 4 \times 10^{-58}$, $\phi_C = 0.36$ (0.33, 0.40)).

Do the differences between analysis methods arise due to differences in how they correct for multiple comparisons across

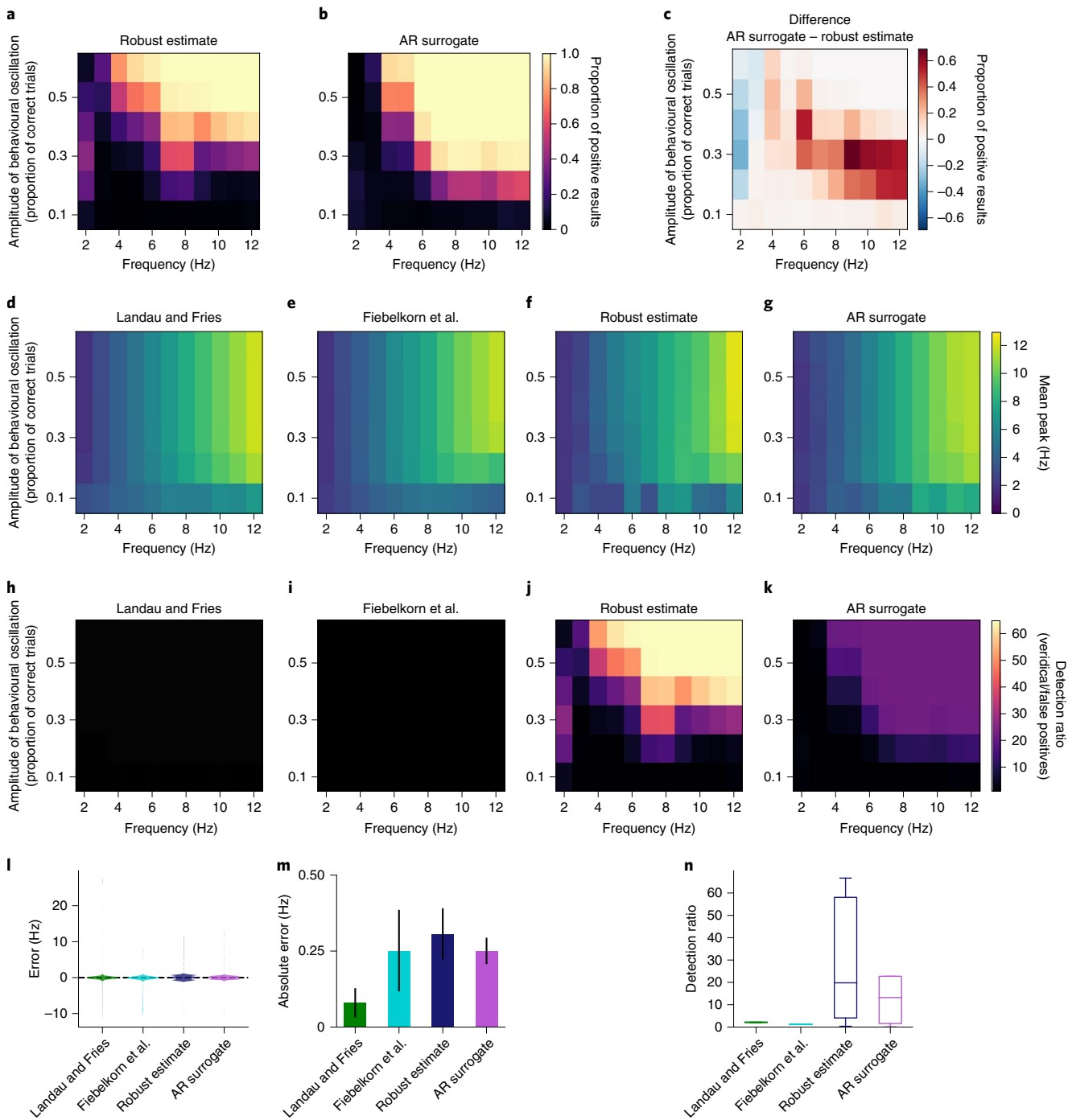

**Fig. 4 | The AR surrogate and robust estimate methods recover oscillations in simulated behaviour. a**, Proportion of simulated experiments that correctly recover an oscillation when analysed with the AR surrogate method, plotted as a function of the frequency and amplitude of behavioural oscillations. The colours correspond to the proportion of experiments that found significant oscillations in behaviour at $P < 0.05$. Each cell shows the results from 1,000 simulated experiments. **b**, As in **a**, but analysed with the robust estimate method. **c**, Difference in the proportion of recovered oscillations between the AR surrogate and robust estimate methods. Positive values correspond to higher experimental power for the AR surrogate method, and negative values to higher experimental power for the robust estimate method. **d**–**g**, The frequency of reconstructed oscillations is highly accurate across all analysis methods. The mean frequency of recovered oscillations is shown separately for each analysis method: the Landau and Fries method (**d**), the Fiebelkorn et al. method (**e**), the robust estimate method (**f**) and the AR surrogate method (**g**). **h**–**k**, Ratio of correct positive results to false positives. The detection ratio is computed as the proportion of significant results when the simulated data include oscillations, divided by the proportion of significant false positives when the data do not include oscillations. The ratio is plotted as a function of the frequency and amplitude of simulated behavioural oscillations, shown separately when analysed using the Landau and Fries method (**h**), the Fiebelkorn et al. method (**i**), the robust estimate method (**j**) and the AR surrogate method (**k**). **l**, Violin plots of the distribution of errors of reconstructed frequencies for each analysis method. **m**, The absolute value of the error of reconstructed frequencies for each analysis method, averaged over each cell of the simulations. The error bars show 95% confidence intervals. **n**, Detection ratio for each analysis method. The centre line indicates the median, the box limits indicate the upper and lower quartiles, the whiskers indicate 1.5 × the interquartile range up to the minimum and maximum, and the points indicate outliers.

**Table 2 | Statistical tests comparing the detection ratios of the different analysis methods**

| Comparison | Estimate | 95% confidence interval | t | P value |
|---|---|---|---|---|
| Landau and Fries, Fiebelkorn et al. | 0.95 | 0.93, 0.96 | 132.81 | $1.3 \times 10^{-110}$ |
| Landau and Fries, robust estimate | 34.04 | 28.37, 39.72 | 11.91 | $1.3 \times 10^{-20}$ |
| Landau and Fries, AR surrogate | −13.94 | −15.86, −12.02 | −14.42 | $9.4 \times 10^{-26}$ |
| Fiebelkorn et al., robust estimate | 34.99 | 29.31, 40.67 | 12.23 | $2.8 \times 10^{-21}$ |
| Fiebelkorn et al., AR surrogate | −14.88 | −16.81, −12.96 | −15.36 | $1.3 \times 10^{-27}$ |
| Robust estimate, AR surrogate | 20.11 | 15.18, 25.03 | 8.11 | $1.7 \times 10^{-12}$ |

Each row shows the results of a linear regression comparing two analysis methods (for example, AR surrogate versus robust estimate). The regressions include terms to control for the amplitude and frequency of the simulated oscillation, with degrees of freedom (3, 96).

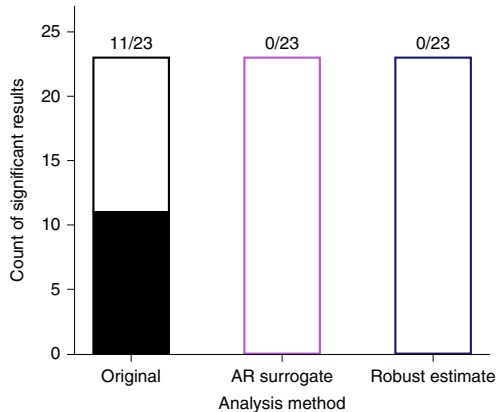

**Fig. 5 | No evidence for behavioural oscillations in reanalyses of published datasets.** Reanalysis of publicly available data from published studies. Each bar shows the number of significant results (filled portion of the bar) and the number of non-significant results (unfilled portion of the bar). The 'Original' bar shows the statistical tests reported in the original studies (all based on shuffling in time), and the 'AR surrogate' and 'robust estimate' bars show the results after reanalysing those data with the alternative analysis methods. The numbers at the top of each bar show (the number of significant results)/(the total number of statistical tests).

frequencies? For the robust estimate analysis (Fig. 6b), the choice of Bonferroni or FDR correction[39] does not significantly influence the rate of either false positives ($\chi^2(1) = 0.2$, $P = 0.7$, $\phi_C = 0.01$ (0.00, 0.06)) or true positives ($\chi^2(1) = 0.5$, $P = 0.5$, $\phi_C = 0.02$ (0.00, 0.06)). For the AR surrogate analysis (Fig. 6c), the rate of false positives depends on the correction method ($\chi^2(2) = 14.0$, $P = 0.0009$, $\phi_C = 0.07$ (0.04, 0.10)) and is adequately controlled with cluster-based permutation tests (0.04, $P = 0.8$) but not with Bonferroni (0.08, $P = 0.00003$) or FDR correction (0.08, $P = 0.00002$). The rate of true positives with the AR surrogate analysis also depends on the correction method ($\chi^2(2) = 55.0$, $P = 1 \times 10^{-12}$, $\phi_C = 0.14$ (0.10, 0.17)), with cluster tests showing slightly lower sensitivity than Bonferroni correction ($\chi^2(1) = 34.0$, $P = 6 \times 10^{-9}$, $\phi_C = 0.13$ (0.10, 0.16)) or FDR correction ($\chi^2(1) = 26.1$, $P = 3 \times 10^{-7}$, $\phi_C = 0.12$ (0.08, 0.15)).

The reliability of these tests depends on the design of the behavioural experiments. Landau and Fries analysed the spectrum of a behavioural time series that was 0.85 s long, sampled at 60 Hz. These simulations show the results of behavioural time series from 1/2 to 2 times this duration. When the data are analysed as in Landau and Fries, false positives are above 0.05 regardless of signal length (Fig. 6d, all > 0.4, all $P < 10^{-259}$). For the robust estimate analysis, the rate of false positives rises slightly with longer time series (Fig. 6e, 0.42 versus 1.7 s: $\chi^2(1) = 13.2$, $P = 0.0003$, $\phi_C = 0.09$ (0.05, 0.12)) but remains below 0.05 for all signal lengths. The rate of true positives, however, rises dramatically with longer time series (0.42 versus 1.7 s: $\chi^2(1) = 947.5$, $P = 5 \times 10^{-208}$, $\phi_C = 0.69$ (0.66, 0.72)). For the AR surrogate analysis, the rate of false positives rises with longer signals (Fig. 6f, 0.42 versus 1.7 s: $\chi^2(1) = 144.8$, $P = 2 \times 10^{-33}$, $\phi_C = 0.27$ (0.23, 0.31)) and is no longer adequately controlled for signals that are twice as long as that used in Landau and Fries (0.22, $P = 10^{-77}$).

Changing the sampling rate of the behavioural time series has a similar effect as changing its length. When the data are analysed as in Landau and Fries, false positives are above 0.05 regardless of the sampling rate (Fig. 6g, all > 0.4, all $P < 10^{-259}$). For the robust estimate analysis, higher sampling rates do not alter the rate of false positives (Fig. 6h, 30 versus 120 Hz: $\chi^2(1) = 0.0$, $P = 0.9$, $\phi_C = 0.01$ (0.00, 0.05)), and they increase the sensitivity of the test to true oscillations (30 versus 120 Hz: $\chi^2(1) = 23.6$, $P = 1 \times 10^{-6}$, $\phi_C = 0.11$ (0.06, 0.15)). For the AR surrogate analysis, the false positive rate rises with increasing sampling rate (Fig. 6i, $\chi^2(1) = 214.4$, $P = 2 \times 10^{-48}$,

$\phi_C = 0.33$ (0.30, 0.36)) and is no longer adequately controlled when behaviour is sampled at 120 Hz (0.21, $P = 10^{-68}$).

The AR surrogate method shows higher sensitivity to true oscillations than the robust estimate method when a behavioural time series has a small number of samples ($\chi^2(1) = 732.3$, $P = 3 \times 10^{-161}$, $\phi_C = 0.61$ (0.57, 0.64)) or when it is measured at a low sampling rate ($\chi^2(1) = 191.3$, $P = 2 \times 10^{-43}$, $\phi_C = 0.31$ (0.27, 0.35)). However, the robust estimate is the only method that appropriately controls the rate of false positives when a behavioural time series has a large number of samples or when it is measured at a high sampling rate. Taken together, these results suggest that, when designing a new study of behavioural oscillations, researchers should simulate a variety of signals to select the best behavioural paradigm and analysis method for the question at hand.

## Discussion

What is the temporal structure of attention? Although the field is approaching consensus that attention moves rhythmically, the computational simulations presented here suggest that this conclusion may be premature. Shuffling in time tests the null hypothesis that a behavioural time course shows no temporal structure whatsoever. This type of analysis therefore yields positive results whenever the data show any structure in time, regardless of whether that structure is oscillatory. In contrast, the AR surrogate and robust estimate methods test the null hypothesis that the data are generated by an AR(1) process, allowing us to disentangle periodic and aperiodic structure. When applied to published datasets, these alternative analyses do not find any evidence for oscillations in behaviour. These findings do not rule out the possibility of there being rhythms in attention. Instead, they show that the current evidence arguing for oscillations cannot distinguish between periodic and aperiodic temporal structure. The present simulations are consistent, however, with the now-widespread conclusion that attention is not sustained uniformly over time[6,7]. These results encourage us to question whether attention switches rhythmically after all.

These results also provide guidance on how future studies could be designed. For example, many researchers are interested in behavioural rhythms around 4 Hz. When using common experimental designs, however, no existing analysis method can reliably

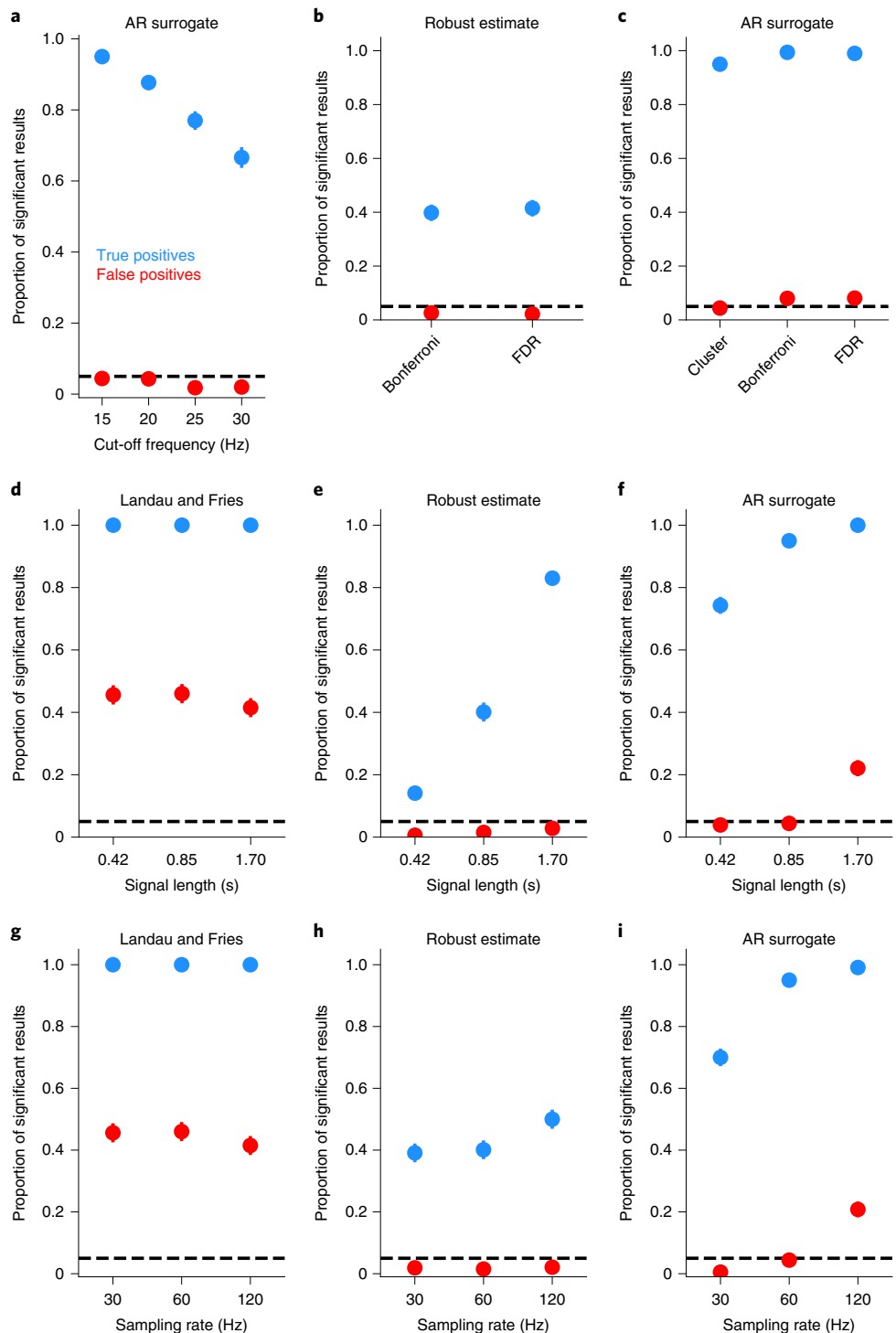

**Fig. 6 | False positives and sensitivity with different analysis choices and experimental designs. a–i**, The proportion of significant positive results for each condition. False positives (red) are computed as the proportion of significant results for data generated as a random walk. True positives (blue) are computed as the proportion of significant results for data generated as a random walk plus an oscillation (frequency 6 Hz, amplitude 0.4, plus a random walk). $k = 1{,}000$ simulated experiments per condition. The error bars show 95% confidence intervals. The dashed lines show the expected rate of false positives ($\alpha = 0.05$). Panel **a** shows the AR surrogate analysis when varying the frequency cut-off. Panel **b** shows the effects of different methods of multiple comparison correction on the robust estimate analysis. Panel **c** shows the effects of different methods of multiple comparison correction on the AR surrogate analysis. Panels **d–f** show the effect of varying the signal length on the Landau and Fries analysis (**d**), the robust estimate analysis (**e**) and the AR surrogate analysis (**f**). Panels **g–i** show the effect of varying the sampling rate on the Landau and Fries analysis (**g**), the robust estimate analysis (**h**) and the AR surrogate analysis (**i**).

distinguish weak 4-Hz oscillations from aperiodic noise (Fig. 4h–k). Future studies of theta-band rhythms in behaviour could solve this problem by extending the length of the behavioural time series and analysing the results using the robust estimate method (Fig. 6e).

When we shuffle the data in time, we test the null hypothesis that the data show no structure in time whatsoever. Any structure in time can therefore lead to significant results in those tests. For example, shuffling in time gives a positive result if the data show

either consistency over trials (for example, accuracy tends to be lower immediately following a cue stimulus) or autocorrelation (for example, accuracy at 300 ms is more similar to accuracy at 333 ms than to accuracy at 1,000 ms). Consistency over trials can also be thought of as 'phase-locking' of the behavioural response; this is orthogonal, however, to the question of whether behaviour is rhythmic. Shuffling in time therefore does not uniquely identify oscillations—instead, it tests whether the data show any kind of temporal structure.

In contrast, the two alternative methods specifically account for non-oscillatory temporal structure. The AR surrogate method fits a model of the data that captures first-order autocorrelational structure, and uses this model to generate a surrogate distribution. This method therefore tests the null hypothesis that the data follow a non-oscillatory 'red noise' pattern. The AR(1) model in this method is agnostic about the underlying neural processes and is not intended to fully capture the temporal structure of behaviour. Instead, it provides a clear null hypothesis against which we can test for oscillations. The robust estimate method was designed to test for rhythms in climate data, and it fits the spectrum to an analytic AR(1) spectrum. Oscillations in climate data are embedded in autocorrelated noise, similar to theorized oscillations in behavioural data. This method therefore also tests the null hypothesis that the data follow a non-oscillatory red noise pattern.

Shuffling in time leads to spectral peaks that could reflect either periodic or aperiodic regularities in behaviour. This finding does not invalidate the literature on attentional switching. On the contrary, it encourages us to consider these results in the context of the rich aperiodic temporal structure in perception and cognition. On short timescales, for example, attention is impaired when two target events appear within around 100–500 ms of each other; this is called the 'attentional blink'[40]. Furthermore, when people search through a visual scene, attention shows 'inhibition of return', with a suppression of perceptual processing of objects that have recently been attended to; inhibition of return appears as quickly as 50 ms after a cue[41], and its effects can last for up to 3 s (ref. [42]). At longer timescales (seconds to hours), behaviour is correlated with itself over time, roughly following a $1/f$ spectrum. This $1/f$ pattern appears in reaction times, accuracy and many other aspects of behaviour[35,36]. Furthermore, rhythmic behaviours can arise without underlying neural oscillations. For example, saccades occur fairly periodically, but saccade timing can be explained using non-oscillatory first-order properties (such as transient inhibition and rebound)[43].

Aperiodic dynamics are pervasive in neural recordings as well as in behaviour[44], and a number of different mathematical approaches have been developed to distinguish between oscillatory and non-oscillatory features of neural time series[45–47]. The $1/f$ slope of a neural power spectrum correlates with the excitation/inhibition balance[48] and predicts a range of different behavioural variables[47,49]. Aperiodic dynamics also appear in phenomena that are widely considered to be inherently oscillatory. For example, the power of high-frequency oscillations often depends on the phase of lower-frequency oscillations[50]. A similar pattern emerges in non-oscillatory scale-free activity: the power of higher-frequency neural activity depends on the 'phase' of aperiodic lower-frequency activity[51]. For a second example, when spiking accompanies a consistent neural oscillation, the information carried by an action potential can depend on the oscillatory phase at which that action potential occurs[52]. This phenomenon is not limited to consistent neural oscillations—the information carried by a spike can also depend on the 'phase' of aperiodic low-frequency activity[53]. This type of non-oscillatory phase coding has been observed in behaving animals. Bats do not show any low-frequency oscillations in the hippocampal formation; but despite this lack of oscillations, hippocampal spiking locks to broadband fluctuations in the local field potential, and spike timing shows non-oscillatory 'phase

precession' as the animal moves through space[54]. The findings summarized above suggest that non-oscillatory dynamics may play an important role in generating temporal structure in the brain and in behaviour[55,56], complementing the well-documented roles of neural oscillations.

The AR surrogate method was designed to test whether a time series shows stronger evidence for oscillations than would be expected from an AR(1) process. This method could be adapted to test a number of related questions and hypotheses. For example, the AR(1) model could be replaced with a $1/f^\beta$ model to test whether a time series shows stronger oscillations than would be expected from a power-law process. Alternatively, fitting the data to an ARMA model could help test for oscillations in the presence of time-lagged errors. This framework can be adapted to use any generative time-series model, allowing researchers to discriminate oscillations from different varieties of aperiodic temporal structure.

Although the current study focuses on the dynamics of attentional switching, the AR surrogate method could be used to identify rhythms in any brief time series. For example, this method could be used to test for other behavioural rhythms, such as in perceptual sensitivity[29] or visual categorization[28]. This method may also be useful for identifying bursts of neural oscillations in ongoing non-oscillatory activity[57,58], by testing whether brief snippets of neural recordings show stronger oscillations than would be expected from the autocorrelated background activity alone. In this case, the generative model of aperiodic activity could be derived from a much longer segment of data, yielding improved estimates of rhythmic activity. Finally, the AR surrogate method could be applied to time series in climate science that are too brief for the robust estimate method to provide reliable results (Fig. 6e).

Attentional switching is not the only aspect of perception that has been proposed to oscillate. A related literature shows robust evidence for rhythmic fluctuations in perceptual sensitivity[2]. Oscillations in perceptual sensitivity have been widely reported in behavioural studies[59,60]. Electrophysiological studies show that perceptual sensitivity depends on the phase of ongoing neural oscillations[18,61–68] (but see ref. [69]). These studies of rhythms in sensitivity do not test for significant oscillations by shuffling the data in time and are therefore not subject to the same statistical issues as the studies of attentional switching.

## Methods

In this study, data from four published studies were reanalysed. These prior data were collected in compliance with local ethical regulations, with experimental procedures approved by the Human Research Ethics Committees of the University of Sydney[14], the Monash University Human Research and Ethics Committee[17], the CERES (Conseil d'Évaluation Éthique pour les Recherches En Santé) ethics committee of Paris Descartes University[21], and the ethics committee of the faculty of psychology and sports science, University of Muenster (no. 2018-36-RM)[25].

In addition, computational simulations of behavioural experiments were analysed according to standard procedures from highly cited papers. The simulations and analyses followed the details from two prominent studies: Landau and Fries[6], and Fiebelkorn et al.[7].

**Simulated behavioural experiments.** Behavioural studies were simulated using the details of the experiments in Landau and Fries, and Fiebelkorn et al. In both experiments, the participants were first presented with visual stimuli on the screen. After a short delay, the participants saw a cue stimulus intended to attract spatial attention and reset ongoing cortical dynamics. After a variable delay, a faint target stimulus appeared briefly at either the cued location or an uncued location. These studies then considered changes in accuracy as a function of the delay between the cue and the target.

In these simulations, each experiment began with an idealized accuracy time course. This is the time course of accuracy that would be obtained after running an infinite number of trials. For each trial, a cue–target delay was randomly selected, with a balanced number of trials at each delay. Accuracy for each trial was randomly determined as a function of the idealized accuracy time course. For example, if a trial was selected for a cue–target delay of 0.5 s, and the idealized accuracy at 0.5 s was 60%, then that trial had a 60% chance of being a hit and a 40% chance of being a miss.

*Experiment details: Landau and Fries.* For simulations following Landau and Fries, time courses were simulated with cue–target delays of 0.15 s to 1.0 s, sampled at 60 Hz. I simulated data from 16 subjects, with each subject having 104 trials at each location.

*Experiment details: Fiebelkorn et al.* For the simulations following Fiebelkorn et al., time courses were simulated with cue–target delays of 0.3 to 1.1 s, sampled at 60 Hz. I simulated data from 15 subjects, with each subject having 441 trials at each location.

*Experiment details: AR surrogate and robust estimate analysis.* For the simulations using the other two methods (AR surrogate and robust estimate), the experiments were simulated as in Landau and Fries.

**Identifying rhythms in simulated behaviour.** I simulated behavioural experiments following the analysis pipelines from Landau and Fries, and Fiebelkorn et al. In addition, experiments were simulated using a permutation method that uses autoregressive models to generate a surrogate distribution. Finally, experiments were simulated using a method for determining spectral peaks in climatic time series[33]. All analyses only considered frequencies below 15 Hz (12 Hz in the Fiebelkorn et al. analysis, reflecting the plotted data).

An experiment is counted as a positive result if it shows any significant peaks in the spectrum after correcting for multiple comparisons across frequencies. For simulations without an oscillatory component, positive results are referred to as 'false positives'.

*Shuffling in time: Landau and Fries.* For this analysis method, I search for rhythms in behaviour following Landau and Fries. To derive the spectrum, accuracy is first averaged over trials and subjects at every cue–target delay. The data are then linearly detrended, tapered with a Hanning window and padded to 256 samples. Finally, the amplitude spectrum is obtained by taking the magnitude of a DFT of this time series.

A randomization procedure is used to determine the statistical significance of peaks in the spectrum. For $k = 500$ permutations, the time of the cue–target delay is randomly shuffled among all the trials, and the spectrum is calculated following the same procedure as used in the empirical data. $P$ values in each simulated experiment are computed as the proportion of values at each frequency that have greater spectral magnitude than the empirical spectrum (a one-tailed test), followed by Bonferroni correction for multiple comparisons across frequencies. For frequencies at which no permutations are stronger than the empirical value, the $P$ value is taken as $P = 0$. Bonferroni correction has no effect on these $P$ values (because zero times any number equals zero). But if one permutation had a higher value than empirical value, then $P = 1/500 = 0.002$, with an adjusted $P$ value of $P = 0.002 \times 256$ samples $= 0.512$. This analysis therefore selects any frequencies at which the empirical value is stronger than every permutation, and excludes all other frequencies.

*Shuffling in time: Fiebelkorn et al.* This analysis method follows the procedures in Fiebelkorn et al. To derive the spectrum, accuracy is first averaged over trials and subjects in overlapping time windows with width 0.05 s, advancing by steps of 0.01 s. This accuracy time series is then detrended with a second-order polynomial, tapered with a Hanning window and padded to 128 samples. Finally, the spectrum is computed by taking the magnitude of the DFT of this time series. To match the plots in Fiebelkorn et al., these analyses only retained frequencies less than or equal to 12 Hz.

Statistical significance is determined as in Landau and Fries, by shuffling the trials in time and recomputing the spectra ($k = 1,000$), correcting over multiple comparisons across frequencies using the FDR[39].

*Robust estimation of background noise.* In addition to the randomization analyses popular in cognitive neuroscience, I used a technique that is common in geology and climate science[33]. This procedure was developed to identify rhythms in geological time series and to isolate these rhythms from a background of autocorrelated noise. I apply this analysis to the time course of average accuracy at each cue–target delay. The mean of the time course is subtracted, and the spectrum is computed using Thomson's multi-taper procedure[70]. For appropriate spectral smoothing, I selected a time-bandwidth parameter of 1.5 with two tapers. I then estimate the aperiodic background spectrum by smoothing the multi-taper spectrum with a median filter (width = 7) and using this robustly smoothed spectrum to fit an estimate of an AR(1) spectrum approximating the aperiodic background activity:

$$S(f) = S_0 \frac{1 - \rho^2}{1 - 2\rho \cos \pi(f/f_N) + \rho^2} \qquad (1)$$

where $f$ is the frequency, $f_N$ is the Nyquist frequency, $S_0$ is the average value of the power spectrum and $\rho$ is the AR(1) coefficient. Finally, I test for statistically significant periodic components by taking the ratio of the multi-taper amplitude spectrum against the robust estimate of the AR(1) background spectrum,

separately for each frequency, and comparing this to a $\chi^2$ distribution with degrees of freedom equal to $2 \times$ (number of tapers). These analyses only retained frequencies less than or equal to 15 Hz. To correct for multiple comparisons across frequencies, $P$ values were adjusted with the Bonferroni correction. For further details, see Mann and Lees[33].

*AR surrogate.* The data were also analysed using a method to test for significant oscillations in autocorrelated time series. This method uses AR(1) models to generate a surrogate distribution for non-parametric bootstrap tests. First, accuracy is averaged at each cue–target delay. I then remove the linear trend and fit an autoregressive model with one parameter:

$$X_t = c + \phi X_{t-1} + \epsilon_t \qquad (2)$$

where $X_t$ is the time series at each time point, $c$ is a constant, $\phi$ is the AR parameter and $\epsilon_t$ is white noise. The AR model is fit using exact maximum likelihood with the Kalman filter. This AR(1) model captures the first-order aperiodic temporal structure in the behavioural time series but does not generate consistent oscillations. This fitted AR(1) model is then used to generate a surrogate distribution of time courses with the same length, AR parameter and residual variance as the empirical time series ($k = 2,000$). The time courses generated by this model preserve the first-order aperiodic structure of the empirical data but lack any periodic components. The time courses from the empirical data and the AR-generated surrogate signals are then linearly detrended, and the spectra are computed by taking the magnitude of the DFT. No tapering, smoothing or zero-padding was applied. In preliminary analyses, tapering was found to drastically reduce both the power of this analysis and the accuracy of the frequency estimates. These analyses only retained frequencies greater than 0 (DC) and less than or equal to 15 Hz.

I correct for multiple comparisons across frequencies using a one-sided cluster-based permutation test[34] (cluster threshold $\alpha$, 0.05; cluster statistic, summed $z$ score). This procedure tests whether the empirical and the surrogate data come from the same distribution[71]. A significant result therefore indicates that the empirical data are not compatible with an AR(1) process. The specific frequencies of any putative oscillations can be interpreted by visual inspection of the peaks in the spectrum. Samples were included in a cluster if their $z$ values exceeded the one-tailed cluster threshold (for $\alpha = 0.05$, $z_{threshold} = +1.64$). A one-tailed threshold was chosen to focus on points at which the empirical spectrum exceeds the surrogate distribution. For each run (the empirical data and each surrogate run), the cluster statistic was computed as the summed $z$ score within each cluster. $P$ values were calculated as the proportion of surrogate runs in which the maximum cluster statistic was greater than or equal to the maximum cluster statistic in the empirical data.

**Simulated behavioural time courses.** To determine how the different analysis methods respond to different types of temporal structure in behaviour, I simulated literatures of 1,000 experiments for each combination of analysis method and temporal structure. Each simulated experiment began with an idealized accuracy time course that was the same across all participants and trials within that experiment. This idealized accuracy formalizes the temporal structure of attentional switching after the cue stimulus. For each trial, a cue–target delay was randomly selected. I then randomly determined whether that trial was a hit or a miss on the basis of the idealized accuracy for that cue–target delay. For example, if one simulation had an idealized accuracy of 70% at a cue–target delay of 0.60 s, then each trial for that time point was determined as a weighted coin toss, with $P(\text{hit}) = 0.7$ and $P(\text{miss}) = 1 - 0.7 = 0.3$.

I tested four types of aperiodic temporal structure. 'Fully random' simulations contained no temporal structure at all, with no consistency over trials. These time series were simulated with an idealized accuracy time course with $P(\text{hit}) = 0.5$ at every cue–target delay. For 'white noise' simulations, idealized accuracy was generated by sampling a random Gaussian process. 'Random walk' simulations were generated with a Gaussian random walk; this is equivalent to a power-law spectrum with an exponent of 2 ($1/f^2$). 'AR(1) noise' simulations were generated with a one-coefficient Gaussian autoregressive process with $\beta = 0.5$. For simulations that included consistency across trials (all except 'fully random'), the idealized accuracy was rescaled to approximate the accuracy range in the behavioural literature: (0.5, 0.7).

For each method and noise type, I tested for an inflated rate of false positives using a one-tailed $t$ test. One-tailed tests were chosen to focus on control of the type I error rate.

Next, I tested how the different analyses reconstruct true oscillations in behaviour. For these simulations, the idealized accuracy time course was generated as a sine wave with randomized phase, at frequencies from 2 to 12 Hz in steps of 1 Hz. The amplitude of these behavioural oscillations (corresponding to the range between minimum and maximum accuracy) was varied from 0.1 to 0.6 in steps of 0.1. Mean accuracy was held at 0.5. These oscillations were then added to a random walk generated as described above, and the resulting time series was used as an idealized accuracy time course.

To measure the precision of each analysis method, I computed the error in recovered rhythms. For experiments with a statistically significant peak after

controlling for multiple comparisons, I identified the frequency with the maximum spectral amplitude.

To quantify the selectivity of each analysis method, I computed a detection ratio: (the rate of correct positive results when an oscillation is present)/(the rate of false positive results when no oscillation is present). Linear regressions were then used to test for differences between each pair of analysis methods. The detection ratio was modelled as a function of the analysis method, with terms to control for the frequency and amplitude of the simulated oscillation. I report the regression coefficient reflecting the analysis method, along with 95% confidence intervals, $t$ statistics and $P$ values.

**Reanalysis of publicly available data.** Data from four previous studies were retrieved from public repositories[14,17,21,25]. Data for Fiebelkorn et al. were requested from the first author, but these data were not provided. Data for Landau and Fries were provided by the authors and analysed with their input. However, we could not exactly reproduce the raw time series plotted in Fig. 2a,c of Landau and Fries. The differences between our replotted data and the original data were fairly small, but the reanalysed data had been preprocessed with a low-pass filter (that is, it had been smoothed in time). Because the alternative methods both rely on fitting an AR(1) model to the data, these smoothed time courses would provide an incorrect estimate of the AR(1) parameter. The statistical conclusions of these analyses would therefore be difficult to interpret. As a consequence, these data were not reanalysed using the AR surrogate and robust estimate methods.

In the publicly available data, every test using shuffling in time was reanalysed using the two alternative methods. For each test, I reproduced the aggregated behavioural time course and verified it against the plot of the time course in the original paper. These time series were then reanalysed using the AR surrogate and robust estimate methods. I counted the number of statistically significant results ($P < 0.05$ after correcting for multiple comparisons) reported in the published paper and compared this with the count of statistically significant results from the alternative methods.

**Variations to the analysis methods.** The simulations reported in Fig. 6 test how altering the analysis methods and experimental design can influence the results ($k = 1{,}000$ simulations per condition). To test for false positives, behaviour was simulated as a random walk. To test for true positives, behaviour was simulated as a random walk plus an oscillation (6 Hz, amplitude 0.4). To test whether the false positive rate was controlled in each condition, the proportion of false positives was compared against $\alpha = 0.05$ using one-tailed binomial tests. Multiple conditions were compared using chi-squared tests, which included the counts of significant versus non-significant results in each condition. Effect size was calculated as Cramér's $V$ ($\phi_C$) with bootstrapped 95% confidence intervals.

To test the effect of cut-off frequency in the AR surrogate analysis, the cut-off frequency was varied between 15 and 30 Hz in steps of 5 Hz.

To test for differences between methods of multiple comparison correction, the AR surrogate analysis was performed using (1) the cluster-based permutation test described above, (2) Bonferroni corrections and (3) FDR correction[39]. The robust estimate analysis was tested with Bonferroni and FDR corrections. The cluster-based test was not examined for the robust estimate analysis, because this analysis does not involve a surrogate distribution that can be randomly permuted.

To test for an effect of the length of the time series, behaviour was simulated according to Landau and Fries (0.85 s), at half this length (0.42 s) and at twice this length (1.7 s). These simulations compared the Landau and Fries, robust estimate, and AR surrogate analyses. The Fiebelkorn et al. analysis was not analysed here because it has uniformly higher rates of false positives than the Landau and Fries method. These simulations maintained roughly the same number of trials within an experiment; this allows us to consider these as different options available to an experimenter, without requiring the experimenter to double the resources required to collect a dataset. To preserve the same number of observations at each time point, the number of total trials differed slightly between conditions (1,647 to 1,664 trials).

To test for an effect of the sampling rate of the behavioural time series, behaviour was simulated according to Landau and Fries (60 Hz), at half this rate (30 Hz) and at twice this rate (120 Hz). As with the simulations varying the length of the time series, these simulations maintained roughly the same number of trials within an experiment. To preserve the same number of observations at each time point, the number of total trials differed slightly between conditions (1,647 to 1,664 trials).

**Reporting summary.** Further information on research design is available in the Nature Research Reporting Summary linked to this article.

## Data availability

The results of all simulations are available at this repository: https://osf.io/6bs4e/. The data used for the reanalyses are publicly available at the following repositories: Ho et al.[14], https://ars.els-cdn.com/content/image/1-s2.0-S09609822173132 09-mmc2.xlsx; Davidson et al.[17], https://figshare.com/projects/Crossmodal_binocular_rivalry_attention_sampling_project/56252; Senoussi et al.[21], https://osf.io/2d9sc/?view_only=6ef3f85d9f944d27b23fc7af5a26f087; and Michel et al.[25], https://osf.io/de4bu/.

## Code availability

All code used to perform the analyses and generate the plots is available at https://github.com/gbrookshire/simulated_rhythmic_sampling.

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

## Acknowledgements

This project has benefited from helpful discussions with O. Jensen, A. Landau, I. Fiebelkorn, P. Fries, F. van Ede, H. Nusbaum, A. Pesquita and the Neuronal Oscillations laboratory at the University of Birmingham. I thank the authors of the reanalysed datasets for making their data publicly available: H. T. Ho, J. Leung, D. C. Burr, D. Alais, M. C. Morrone, M. J. Davidson, J. J. A. van Boxtel, N. Tsuchiya, M. Senoussi, J. C. Moreland, N. A. Busch, L. Dugué and R. Michel. This study was supported by a Wellcome Trust Investigator Award in Science (no. 207550) awarded to O. Jensen. The funders had no role in study design, data collection and analysis, decision to publish or preparation of the manuscript.

## Author contributions

G.B. formulated the research questions, performed the analysis and wrote the paper.

## Competing interests

The author declares no competing interests.

## Additional information

**Correspondence and requests for materials** should be addressed to Geoffrey Brookshire.

# Reporting Summary

## Statistics

For all statistical analyses, confirm that the following items are present in the figure legend, table legend, main text, or Methods section.

| n/a | Confirmed | |
|---|---|---|
| ☐ | ☒ | The exact sample size (n) for each experimental group/condition, given as a discrete number and unit of measurement |
| ☐ | ☒ | A statement on whether measurements were taken from distinct samples or whether the same sample was measured repeatedly |
| ☐ | ☒ | The statistical test(s) used AND whether they are one- or two-sided *Only common tests should be described solely by name; describe more complex techniques in the Methods section.* |
| ☐ | ☒ | A description of all covariates tested |
| ☐ | ☒ | A description of any assumptions or corrections, such as tests of normality and adjustment for multiple comparisons |
| ☐ | ☒ | A full description of the statistical parameters including central tendency (e.g. means) or other basic estimates (e.g. regression coefficient) AND variation (e.g. standard deviation) or associated estimates of uncertainty (e.g. confidence intervals) |
| ☐ | ☒ | For null hypothesis testing, the test statistic (e.g. $F$, $t$, $r$) with confidence intervals, effect sizes, degrees of freedom and $P$ value noted *Give P values as exact values whenever suitable.* |
| ☒ | ☐ | For Bayesian analysis, information on the choice of priors and Markov chain Monte Carlo settings |
| ☐ | ☒ | For hierarchical and complex designs, identification of the appropriate level for tests and full reporting of outcomes |
| ☐ | ☒ | Estimates of effect sizes (e.g. Cohen's $d$, Pearson's $r$), indicating how they were calculated |

*Our web collection on statistics for biologists contains articles on many of the points above.*

## Software and code

Policy information about availability of computer code

| Data collection | No new data-sets were collected for this study. Simulated data were generated using the code described below under "Data analysis." |
|---|---|
| Data analysis | All code used to perform these analyses is available online at https://github.com/gbrookshire/simulated_rhythmic_sampling. These analyses were performed using open-source Python code. The exact versions of all external libraries are specified in "requirements.txt" at the Github repository. |

For manuscripts utilizing custom algorithms or software that are central to the research but not yet described in published literature, software must be made available to editors and reviewers. We strongly encourage code deposition in a community repository (e.g. GitHub). See the Nature Portfolio guidelines for submitting code & software for further information.

## Data

Policy information about availability of data

All manuscripts must include a data availability statement. This statement should provide the following information, where applicable:
- Accession codes, unique identifiers, or web links for publicly available datasets
- A description of any restrictions on data availability
- For clinical datasets or third party data, please ensure that the statement adheres to our policy

All simulated data is available at a public repository: https://osf.io/6bs4e/. The paper provides links to all previously-published datasets that were reanalyzed here.

# Field-specific reporting

Please select the one below that is the best fit for your research. If you are not sure, read the appropriate sections before making your selection.

☐ Life sciences  ☒ Behavioural & social sciences  ☐ Ecological, evolutionary & environmental sciences

For a reference copy of the document with all sections, see nature.com/documents/nr-reporting-summary-flat.pdf

# Behavioural & social sciences study design

All studies must disclose on these points even when the disclosure is negative.

| | |
|---|---|
| Study description | Quantitative study based on computational simulations and reanalysis of publicly-available data. |
| Research sample | We re-analyzed data from four papers.<br>- Ho et al. (2017): "Twenty healthy adults (7 male, 3 left-handed, mean age 21.8 ± 3.9) with normal hearing" from around the University of Sydney.<br>- Davidson et al. (2018): "34 healthy individuals (21 females, 1 left-handed, average age 23 ± 4.7) were recruited via convenience sampling at Monash University, Melbourne, Australia".<br>- Senoussi et al. (2019): "Thirteen human observers (nine women, four men; age [M ± SD] = 20.9 ± 0.8 years; range: 20–22)" around Paris Descartes University.<br>- Michel et al. (2021): "Fourteen participants participated in the main study (10 women, 13 right-handed, 11 right-eye dominant, aged 18–28 years, Mage = 21.4, SDage = 2.6)." Participants were recruited from the University of Muenster. |
| Sampling strategy | Reanalyzed data-sets used convenience samples, with sample sizes based on previous research in the literature. |
| Data collection | Reanalyzed data-sets were collected on computers. The experimental conditions were manipulated on a trial-by-trial basis, and by virtue of the design any experimenter would have been blind to the relevant experimental condition (i.e. the delay between the cue and the target stimulus). |
| Timing | The reanalyzed data-sets do not specify the dates of data collection. |
| Data exclusions | I reanalyzed the data as provided by the original study authors. No additional data were excluded from the analyses.<br>- Ho et al. (2017): None specified.<br>- Davidson et al. (2018): None specified.<br>- Senoussi et al. (2019): "Due to technical issues during data recording, two observers were excluded from the analysis."<br>- Michel et al. (2021): "An additional participant did not complete the preregistered minimum number of sessions and was therefore excluded. One participant had previously participated in the pilot experiment." |
| Non-participation | - Ho et al. (2017): None specified.<br>- Davidson et al. (2018): None specified.<br>- Senoussi et al. (2019): None specified.<br>- Michel et al. (2021): "An additional participant did not complete the preregistered minimum number of sessions and was therefore excluded." |
| Randomization | The experimental conditions were varied within-subjects on a trial-by-trial basis. |

# Reporting for specific materials, systems and methods

We require information from authors about some types of materials, experimental systems and methods used in many studies. Here, indicate whether each material, system or method listed is relevant to your study. If you are not sure if a list item applies to your research, read the appropriate section before selecting a response.

## Materials & experimental systems

| n/a | Involved in the study |
|---|---|
| ☒ | ☐ Antibodies |
| ☒ | ☐ Eukaryotic cell lines |
| ☒ | ☐ Palaeontology and archaeology |
| ☒ | ☐ Animals and other organisms |
| ☐ | ☒ Human research participants |
| ☒ | ☐ Clinical data |
| ☒ | ☐ Dual use research of concern |

## Methods

| n/a | Involved in the study |
|---|---|
| ☒ | ☐ ChIP-seq |
| ☒ | ☐ Flow cytometry |
| ☒ | ☐ MRI-based neuroimaging |

# Human research participants

Policy information about studies involving human research participants

| | |
|---|---|
| Population characteristics | See above. |
| Recruitment | Convenience samples. All manipulations were performed within-subjects, so it is not likely that biases in recruiting substantially changed the results. |
| Ethics oversight | - Ho et al. (2017): "The study was approved by the Human Research Ethics Committees of the University of Sydney."<br>- Davidson et al. (2018): "Monash University Human Research and Ethics Committee approved this study"<br>- Senoussi et al. (2019): "All procedures were approved by the CERES (Conseil d'Évaluation Éthique pour les Recherches En Santé) ethics committee of Paris Descartes University."<br>- Michel et al. (2021): "approved by the ethics committee of the faculty of psychology and sports science, University of Muenster (#2018-36-RM)" |

Note that full information on the approval of the study protocol must also be provided in the manuscript.

