## [Peer Review File · Nature Human Behaviour]

Peer Review Information

Journal: Nature Human Behaviour

Manuscript Title: Geoffrey Brookshire

Corresponding author name(s): Putative rhythms in attentional switching can be explained by aperiodic temporal structure

Reviewer Comments & Decisions:

Decision Letter, initial version:

26th March 2021

Dear Dr Brookshire,

Many thanks for submitting your manuscript entitled "Re-evaluating rhythmic attentional switching: Spurious oscillations from shuffling-in-time", for consideration at Nature Human Behaviour.

My colleagues and I found this work very interesting. However, we feel that in the current form, the manuscript lacks sufficiently strong evidence to make it highly impactful. Currently, you do not carry out any analysis of experimental data, instead relying on simulations. This means that you are unable to show whether any prior findings in this field are in fact affected by inappropriate analysis methods.

If you were able to extend the manuscript to include analysis of existing datasets (e.g. through data sharing by the original authors), we would be very interested in considering a revised version of this work. Please contact me if you'd like to discuss this decision further or if you have any questions about the kinds of analyses we would be interested in seeing.

Thank you for your interest in Nature Human Behaviour. I am sorry that we cannot be more positive at this time.

Sincerely,
Jamie

Dr Jamie Horder

Senior Editor
Nature Human Behaviour

Although we cannot offer to publish your paper in Nature Human Behaviour, the work may be appropriate for another journal in the Nature Research portfolio. If you wish to explore suitable journals and transfer your manuscript to a journal of your choice, please use our **[REDACTED]**. If you transfer to Nature-branded journals or to the Communications journals, you will not have to re-supply manuscript metadata and files. This link can only be used once and remains active until used. All Nature Research journals are editorially independent, and the decision to consider your manuscript will be taken by their own editorial staff. For more information, please see our http://www.nature.com/authors/author_resources/transfer_manuscripts.html?WT.mc_id=EMI_NPG_1511_AUTHORTRANSF&WT.ec_id=AUTHOR manuscript transfer FAQ page. Note that any decision to opt in to In Review at the original journal is not sent to the receiving journal on transfer. You can opt in to *In Review* at receiving journals that support this service by choosing to modify your manuscript on transfer. In Review is available for primary research manuscript types only.

Decision letter, first revision

1st June 2021

Dear Dr Brookshire,

Thank you once again for your manuscript, entitled "Re-evaluating rhythmic attentional switching: Spurious oscillations from shuffling-in-time," and for your patience during the peer review process.

Your manuscript has now been evaluated by 4 reviewers, whose comments are included at the end of this letter. Our independent reviewers find your work to be of high interest. However, they also raise some important concerns. We are very interested in the possibility of publishing your study in Nature Human Behaviour, but would like to consider your response to these concerns in the form of a revised manuscript before we make a decision on publication.

In our view, the most important concern, expressed especially by Referee #3 as well as by the signed comments (see below), is over the two alternative analysis methods you suggest. Referee #3 makes a number of detailed points as to how the proposed analysis methods could be improved and better validated. We believe that your manuscript will be greatly strengthened if you are able to respond to these comments with a substantial revision which provides robust and well-validated methods that will be of use to future researchers in the field. Please contact me if you wish to discuss these revisions in more detail.

We also solicited signed comments by the authors of the 4 original datasets your work reanalyzes. 2 sets of authors agreed to submit a signed comment, which are attached to this email. Please note that signed comments do not function as reviews. The authors of signed comments were asked to comment solely on methodological aspects of your work, as well as on the extent to which their work is accurately represented in your manuscript. Decisions on whether to reject a manuscript or invite a revision are based only on the feedback of our independent reviewers. However, if a decision to invite a revision is made, signed comments inform our requests for necessary revisions. (For full details on our Signed Comments policy, please see <https://www.nature.com/nathumbehav/signed-comments>.)

In sum, we invite you to revise your manuscript taking into account all reviewer and editor comments. We are committed to providing a fair and constructive peer-review process. Do not hesitate to contact us if there are specific requests from the reviewers that you believe are technically impossible or unlikely to yield a meaningful outcome.

We hope to receive your revised manuscript within 16 weeks (four months). This is longer than our usual revision timeframe as we recognize that the required revisions may require substantial work.

- Include a "Response to the editors, reviewers, and signed comments" document detailing, point-by-point, how you addressed each editor, referee, and signed comment. If no action was taken to address a point, you must provide a compelling argument. This response will be used by the editors to evaluate your revision and sent back to the reviewers along with the revised manuscript.
- Highlight all changes made to your manuscript or provide us with a version that tracks changes.

[REDACTED]

We look forward to seeing the revised manuscript and thank you for the opportunity to review your work. Please do not hesitate to contact me if you have any questions or would like to discuss these revisions further.

Sincerely,

Jamie

Dr Jamie Horder
Senior Editor
Nature Human Behaviour

REVIEWER COMMENTS:

Reviewer #1:
Remarks to the Author:
Summary-

A growing literature claims that attention switches back and forth rhythmically between locations rather than staying constant, and the rhythmicity has important implications for the underlying neural mechanisms. These ideas have become the dominant viewpoint. The present study calls into question this conclusion on the basis of significant problem with the way the data have been analyzed. However, the manuscript goes far beyond a simple critique and makes the following points:

1) The analysis method used to assess the presence of oscillations in previous studies in this area is clearly flawed and cannot distinguish between oscillations and aperiodic types of temporal structure (e.g., a random walk).

The author said it extremely well himself: "...these studies test the null hypothesis that the behavioral data have no structure in time. These tests do not, therefore, provide unique evidence for oscillations in behavior. Instead, they provide evidence for any kind of structure in time."

2) Two alternative methods can be used that do not have this problem.

3) Simulations demonstrate that the previous methods lead to many false positives, but the two new methods do not.

4) The new methods have a much higher ratio of true positives to false positives.

5) When the new methods are applied to 4 key studies that claimed to provide evidence of oscillations in attention, no evidence for oscillations was found.

6) Temporal structure is present in the prior studies, but it is aperiodic rather than oscillatory.

7) Aperiodic temporal structure is a common feature of complex systems, including recurrent neural networks, and they can indicate important and interesting properties of the systems. By quantifying these aperiodic effects (and not confusing them for oscillations), we can gain new insights into the underlying mechanisms of attention.

Critique-

This is the most important manuscript I have reviewed in several years. It will be highly controversial, and it challenges the dearly held beliefs of many researchers. I can imagine that some researchers will find many things to complain about. For example, they might argue that new methods proposed for assessing oscillations are not optimal (even though they are certainly better than the status quo!). See also Point 4 below.

In my view, however, this paper will have an enormous positive impact on the field, making it possible for researchers to distinguish between periodic and aperiodic temporal structure, both of which are important and have significant implications for the underlying mechanisms of attention. It will not be the last word on this topic — instead, it will lead to many new and important studies. But isn't that what we want for a paper in this journal?

I do, however, have some recommendations for improvement:

1) The issue of whether a pattern of behavioral data is truly oscillatory has also been addressed recently by Yuval-Greenberg and her colleagues (Amit, R., Abeles, D., Bar-Gad, I. et al. Temporal dynamics of saccades explained by a self-paced process. *Sci Rep* 7, 886 (2017). <https://doi.org/10.1038/s41598-017-00881-7>). It would be worth looking at this work to see if there are common solutions. (However, if it turns out to be only tangentially related, the author should not feel obliged to cite it.)

Along a similar line, it would be worth pointing out (in the Intro or even the Abstract) that the methods developed in this paper could be applied to other cases in which researchers ask whether an underlying process is periodic.

2) I may have missed it, but I couldn't find a detailed description of the autoregressive model that was used as one of the alternative analysis methods.

3) Near the top of page 6, it should be mentioned that the new AR method could be used with virtually any correction for multiple comparisons, etc. That is, it is important to note that it has just as much flexibility as the current, flawed approach.

4) With regard to the simulations shown in Figure 4: I didn't look up these previous studies, but my guess is that they reported oscillations at higher frequencies than were found in the simulations. I can imagine that other reviewers or readers will say that this invalidates the simulations. However, I assume that the particular peak frequency in these simulations will depend on the nature of the aperiodic signal. We don't know the exact nature of the aperiodic signal, so it's not a problem if this particular set of simulations doesn't produce the sample spectrum that was found in the prior research.

5) On p. 12, where it says "This finding suggests that rhythms in behavior may not be as prevalent as the published literature suggests," I think it would be fairer to say something like "While no analysis can conclusively prove the absence of oscillations, the present results suggest that..."

6) The discussion of related issues in EEG/ERP research was quite thoughtful, but there are some other methods that might also be relevant in the present context. For a method to determine that multiple cycles are actually present, see:

Watrous, A. J., Fried, I., & Ekstrom, A. D. (2011). Behavioral correlates of human hippocampal delta and theta oscillations during navigation. *Journal of Neurophysiology*, 105(4), 1747–1755.
<https://doi.org/10.1152/jn.00921.2010>.

Recent work by Brad Voytek is also relevant, such as:

Donoghue, T., Haller, M., Peterson, E. J., Varma, P., Sebastian, P., Gao, R., Noto, T., Lara, A. H., Wallis, J. D., Knight, R. T., Shestyuk, A., & Voytek, B. (2020). Parameterizing neural power spectra into periodic and aperiodic components. *Nature Neuroscience*, 23(12), 1655–1665.
<https://doi.org/10.1038/s41593-020-00744-x>.

However, the author should feel free to disregard these papers if they turn out not to be very relevant or to fit the narrative.

Steve Luck (signed review)

Reviewer #2:

Remarks to the Author:

In the manuscript entitled "Re-evaluating rhythmic attentional switching: Spurious oscillations from shuffling-in-time" the author claims that the behavioural oscillations could reflect aperiodic dynamics in attention, rather than periodic rhythms.

The author first performs a comprehensive analyses to demonstrate that shuffling in time commonly applied in time series analysis tests the null hypothesis that a time-series contains no temporal structure whatsoever. After shuffling in time the autocorrelation theoretically becomes zero at all non-zero time-lags. Thus shuffling cannot distinguish between periodic and aperiodic temporal structure in behavioural time-series. The previous test the author confronts with another null hypothesis that the data are autocorrelated, but not oscillatory, allowing us to distinct periodic structure from aperiodic structure.

Here, using computational simulations, the author hypothesises that the behavioural oscillations reflect aperiodic dynamics in attention. The author provides tools to distinguish between oscillatory and non-oscillatory structure, applying two procedures. The first novel randomisation method creates the surrogate distribution using an auto-regressive AR model which serves to captures the aperiodic structure. Besides, the method applies a discrete Fourier transform.

The second procedure, called the 'robust estimate' is commonly used to identify rhythms in climate science. This method was developed to identify oscillations in autocorrelated background noise in geological time-series, and therefore may help distinguish behavioural rhythms from aperiodic background activity.

I find the results persuasive and therefore accept the paper. However, I suggest to cite a paper A. Majdandzic et al, "Spontaneous Recovery in Dynamical Networks," Nature Physics 10, 34-38 (2014). Namely, in a rhythmic theory of attention different studies have argued for rhythmic attentional switching between locations at the frequency of the spectral peak. The paper of Majadandzic et al suggested the spontaneous emergence of macroscopic 'phase flipping' phenomena.

Reviewer #3:

Remarks to the Author:

This manuscript questions the methodology used in many articles to evaluate the presence of rhythmic oscillations in attentional switching. Specifically, it convincingly argues that shuffling time points to obtain statistical significance (for a given frequency) merely tests whether the spectra at that given frequency is higher than it would be for random noise. Therefore, one cannot claim that it is a proof of rhythmic oscillations (at this frequency). Indeed, aperiodic signals will also display higher spectrum values than random noise in some frequencies.

This manuscript then proposes two alternative procedures that allegedly circumvent the problem and it provides simulations to make the case.

This manuscript is well written, and the discussion is quite rich, in term of literature review, links with other research questions and interesting reflections.

I fully agree with the author that the shuffling methodology used presently in the literature cannot be used to provided evidence of rhythmic oscillations (it is unfortunately not the only instance in the literature where the shuffling or permutation schemes do not correspond to the intended purpose).

And the fact that the 11 significant results from the literature (out of 23 tests) became all non significant when compared to an (arhythmic) AR(1)-based null distribution is quite convincing that the simplistic shuffling procedure possibly misled important scientific findings.

Although the proposed solutions are clearly better than the actual practice, I'm a little less enthusiastic concerning the two proposed solutions for the following reasons:

- AR(1) is just one possible aperiodic signal out of so many possible alternatives
- AR(1) is stationary and the author acknowledges that attentional switching data are probably often non-stationary.
- AR(1) results are probably very sensitive to the chosen sampling rate. Doesn't AR(1) behave quite differently if one uses only half of the time-points or on the contrary twice as much ? Other approaches based on fitting the spectrum (as mentioned p15) or equivalently on the ACF (see e.g. Abrahamsen) might be more robust to sampling rate choices ?

I encourage the author to propose a more global approach, e.g. by first evaluating (visually or by a test) whether the stationarity assumption is tenable (e.g. by looking at a time-frequency plot like the continuous wavelet transform), second to check with a few more complex models than an AR (1) and third to discuss more on the fact and consequences that other aperiodic models could still modify the results. The above suggestions are just ideas and more sophisticated methods based on the extremely vast literature on time-series might be even more adequate.

Technical issues:

- In the novel approach, a cluster-based permutation test is proposed to tackle the multiple comparison problem. I see however two difficulties here. First, as stated e.g. in the reference [34], when a cluster is significant, one cannot state that each point within the cluster is significant (but only that there is a point within the cluster that is significant). So it would be incorrect in the interpretation and in the simulation to act as if the frequencies within a significant cluster are all significant. Second, the assumptions for the construction of the null distribution of a cluster-based permutation rely on invariance of correlation between equidistant points, i.e. some sort of stationarity. Here the points are frequencies and I do not see reason to believe in stationarity across frequencies of spectral estimations.
- to allow us to better understand if the difference in the results are due to the core procedure or to the subsequent multiple comparison methods, the simulations should as much as possible separate the two: first compare the results when no multiple comparison corrections are applied, second, when possible, use the same multiple comparison corrections and finally compute the results of the full methods.
- the method and the exact simulation settings should be clearer: for example, (1) to compute the rate of false positives do you consider a sample significant as soon as one frequency is significant or is it the average of significance over all frequencies/tests ? (2) if a cluster is significant, how do you treat each point within the cluster ?
- I am surprised by the fact that as stated in bottom of page 18, the author used $k=500$ shufflings. It implies that the smaller raw p-value is $1/500=0.002$. If I understand correctly the settings, a Bonferroni correction on 256 tests is then applied, which means that the smaller possible corrected p-value is $0.002 * 256 = 0.512$ (!), so that no results can be significant ! So first I'm surprised to see significant results in these simulations and second, the number of shufflings should be increased so that the smallest possible corrected p-value is at the very very most 0.025.
- In the simulation results of Fig4a, why are the error rates of AR-based methods close to 0 instead of being close to 5% ? Does it imply they are highly conservative approaches?
- I did not find which multiple correction procedure is proposed (and used in the simulations) for the second proposal.

more minor comments:

p5 "This AR(1) model captures the aperiodic structure – but not the periodic structure – of the time-series". This claim is too strong.

p15 "The alternative analyses proposed here test the null hypothesis that the data are autocorrelated" is slightly an over statement since it does not cover AR(p)

p15 "However, the simulations and analyses reported here are consistent with the now-widespread conclusion that attention is non-stationary ": I don't see how this can be true since the AR(1) is stationary.

p14 I'm not sure what is meant by "lower-frequency aperiodic activity"

p17: "All p-values reflect two-tailed tests": does it mean that frequency values that are lower than the null distribution are deemed significant ?

Abrahamsen P (1997) A Review of Gaussian Random Fields and Correlation Functions
https://www.nr.no/directdownload/917_Rapport.pdf

Reviewer #4:

Remarks to the Author:

In recent years, several influential studies observed that attention samples sensory stimuli rhythmically. The author proposes an important analysis approach to test for veridical rhythmic modulation of attentional sampling. Critically, previous research relied on an analysis that shuffles the data in time in order to generate surrogates to compare the empirical data against. As the author points out, such an approach tests the null hypothesis that the data contain no structure in time at all. However, it cannot distinguish between periodic rhythms and aperiodic temporal structure. In the present article, the author first presents the problem, suggests two alternative analysis approaches and uses simulated data to demonstrate their superiority above two conventional analysis approaches used in the literature, and finally shows that these new analysis approaches do not find evidence for rhythmic modulation of attention in published datasets.

This is an important article, relevant to a large audience. The article is well written and easy to understand. The conclusions drawn are well motivated and supported by the simulated data and re-analyses of published datasets.

Major

While I appreciate the logical structure of the article, there is one obvious question that remains somewhat unaddressed. It is based on the fact that the data simulations test false positive and true positive rates for the analysis approaches of two highly influential studies in the field and compare them to the alternative approaches. The final results section, however, reveals false positives in actual data of four other studies (which are arguably less influential and present generally somewhat weaker evidence for the temporal modulation of attentional sampling). After reading the article, the reader is thus left alone with guessing/wondering whether the results of the two highly influential articles (LF2012, FSK 2013) present false positives or veridical periodic modulation of attentional sampling. In case the authors of LF2012 and FSK 2013 do not agree with sharing their data for a re-analysis in the present article, this should be stated in a very transparent way.

A major problem of published articles on rhythmic attentional sampling is also the fact that the rhythmic modulation is typically observed in a very limited time interval (usually up to 1s). The author might want to demonstrate to what extent (if any) the data simulation results depend on the length of the sampled time window. This would be very helpful for the audience, especially with regard to the design of future experiments.

One approach that is sometimes considered an alternative to 'shuffling in time' is 'shifting in time'. Shifting in time does not remove the temporal structure in the data but instead it abolishes any phase-locked effects. While 'shifting in time' is certainly not always a sensible analysis approach, it can be used in case the phase-locking (and/or periodic relation) of two time-domain signals (e.g., neural

and behavioral sampling of a stimulus) is tested. It would be helpful if the author could refer to this approach as well.

Minor

I was not entirely sure but it seems the analysis code is available in python. In order to make it accessible to a broader audience, it would be great if it would be available for matlab as well.

SIGNED COMMENTS:

Dear Nature Human Behaviour editorial team,

We thank the editors for this opportunity to provide a signed comment. In this paper, a widely-used permutation method which shuffles the temporal order of events to create a null-distribution of time-series data is challenged. The paper deduces that false positives may be high using this method, because other methods (discussed by the author) do not find true positives. The author suggests that empirical peaks in the observed data may actually be due to other, aperiodic sources of temporal structure (such as autocorrelation). A key feature is that temporal shuffling would also destroy these aperiodic temporal structures, not just oscillatory information, so the resulting null-distributions may confound the absence of an oscillation with the absence of aperiodic structures.

The author recommends replacing temporal shuffling with two alternative methods to test for the presence of oscillations in time-series data (with a null hypothesis being that there exists aperiodic structures without any oscillations). The first alternate method (AR(1)) creates surrogate datasets which preserve the auto-regressive nature of the empirical data. The empirical spectrum is compared to these surrogate datasets, and a 1-dimensional cluster test is applied to assess significance. The second alternate method (Robust est.) is borrowed from climate science, and relies on a multi-taper method with fixed parameters, spectral smoothing, and Bonferroni corrections across frequencies.

While we think this is an important paper that will help move the field forward, we have several concerns regarding the interpretation of, specifically, our previously published dataset, and, generally, the sensitivity of the alternate methods proposed.

Our data are tested against these two methods, after first subjecting the raw time-series to the pre-processing pipeline of other papers (LF2012 or FSK2013) which includes mean subtraction and linear detrending, steps we did not originally perform. From the provided code, it appears the empirical peaks are still above the (uncorrected) 95% CI of the AR surrogate analysis. However, these peaks are singular, and thus do not survive the 1-dimensional cluster correction for adjacent significant peaks in the spectrum (*Mismatch*; $f = 3.37$ Hz, $p = .0097$; *Match*; $f = 8.08$ Hz, $p = .0037$). The Robust est. method does not return significant peaks.

Figure 1. Results from AR(1) method. a-c) Our raw time series of the proportion of first switches during binocular rivalry, sampled at 60 Hz. d) Empirical spectra and AR surrogate 95% CI. Dotted lines display the uncorrected 95% CI.

Figure 2. Results from the Robust est. method. Solid lines display our empirical spectra after multi-taper analysis. Dotted lines display the median smoothed spectra. Broken lines display the surrogate background spectra used for significance testing. The very conservative nature of this test is clear, even in comparison to the AR(1) method.

It's important to note that our method/pipeline is different to LF2012 or FSK2013, and was carefully selected due to the highly unique nature of our dataset – unique even in the context of attentional sampling research. We do shuffle in time, but do not detrend, Hanning taper, or otherwise smooth our analysis as is usually performed. We deliberately computed the fit with a single taper to maximise frequency-resolution, and included important negative control conditions to test for the absence (or change) in attentional sampling frequency consistent with theory. More specifically, we report a difference in frequency between conditions (mismatch and match), and absence in strong control conditions (visual only - no cues, attended vs unattended), as well as convergent EEG data. Importantly, we also do not investigate accuracy over time by systematically varying the SOA between a target and probe, as is standard in attentional sampling research. As a result, we do not have an equivalent amount of data at each time-point, and by design, we have a large aperiodic signal in our time-series, as the proportion of first-switches has a heavy positive skew. This clustering of data reduces the estimated likelihood of detecting a sustained oscillation at later time-points, and reduces the sensitivity of our data when subjected to the Robust est. method (more below).

Without supplying the highly unique context of our original design, the author misrepresents the likelihood of the new alternate methods of detecting an oscillation when reanalysing our dataset. This point is clear when investigating the simulations provided in the manuscript (Table 1 and Figure 6). Although the author states that both methods “successfully recover true oscillations in simulated behaviour”, this is only reliably the case at higher frequencies (above 4 Hz), and when the amplitude of oscillations is above 0.3. As we focus on low frequencies (below 4 Hz) and have small oscillations (amplitude < 0.3), these new methods would be unable to recover true oscillations, even in the simulated data.

It is also important to note that the Robust est. method may not be as applicable to oscillations in behaviour - owing to the assumptions of that method that were determined for application in climate science (Figure 2). In particular, the multi-taper method of Mann and Lees (1996) invokes assumptions "that are faithful to our understanding of the physics governing the climate system", such as isolating peaks only with a coherent phase spectrum, and identifying relatively large amplitude oscillations. It is entirely plausible that attentional sampling is quasi-rhythmic, rather than being determined by a clock-like or climatic oscillation, and that the depth of this oscillation tapers with attentional focus. This interpretation of a short lived burst of attentional sampling is consistent with our (and others) data, but would not be detected by the Robust est. method.

Figure 2. a) The unique processing pipeline of the Robust est. method in climate science. b) An example simulated oscillation used in its validation, with relatively large and sustained oscillations typical in climate science. c) Our detrended data for comparison, with a positive skew as switches become less likely to occur over time.

The overly-conservative nature of these tests is also evident in Table 1 in the paper, which displays the proportion of false-positives for each analysis method. A well calibrated method would produce false positives at the expected rate of 0.05. However, both the Robust and AR surrogate methods are appreciably below this cut-off. For unstated reasons, the author only performed one-tailed tests, showing significantly "greater" than .05 in bold. We believe it is only fair to perform two-tailed tests and report their results, revealing the conservative nature of these results. The author is then expected to comment on this property of the alternative methods.

In summary, while we agree that clarification and improved methods are needed to establish true 'oscillations' in behaviour, the new proposed methods, however, appear overly conservative for the depth of oscillations typically reported in our field. Additionally, our study is particularly unique, with features that decrease the likelihood of detecting oscillations with these new methods - as is evident in the authors' own simulations.

These caveats are critical to restate when introducing the reanalysis of our (and other authors' data). The author should state what the expected likelihood of detecting 'true' oscillations is in the reanalysis of datasets - based on the frequency range and depth of oscillations we have reported. When provided with these caveats, it is unsurprising that with

a different fixed preprocessing pipeline, and more conservative methods, oscillations are not recovered.

Signed,

Matt Davidson, Naotsugu Tsuchiya, Jeroen van Boxtel.

Reply to Brookshire

The autoregressive modelling method proposed by the author is typically applied to historical data, such as in climate science, where accurate predictions about the future rely on past measurements. Here it is being applied in a relatively novel way to our binned data, which have been sorted and grouped relative to a reset signal, rather than in chronological order. This is not necessarily wrong, but interpretations are less immediate.

We applied his proposed method to our data and found that all five oscillations reported in Ho et al. (2017) had a goodness of fit (R^2) larger than 95% of the AR(1) simulations at the reported theta and alpha frequencies. The only condition where we obtained a non-significant result when compared against AR(1) noise was also non-significant in the original publication.

Figure 1 shows the analysis applied to the Ho et al. (2017) result that is most relevant to the author's discussion of rhythmic attentional sampling. In this analysis, we compare the difference and sum of sensitivity (measured as d -prime) between the ears (corrected for multiple comparisons with maximal statistics, Ho et al., 2019; Ho et al., 2020). For every AR(1) simulation ($N=1000$), we fitted a sinusoidal curve with frequencies between 4 and 10 Hz in steps of 0.1-Hz and selected the maximal R^2 (measure of goodness of fit), irrespective of frequency, phase and amplitude. We then compared the maximal R^2 of the original time series against the distribution of 1000 maximal R^2 . The significant theta oscillation 5.9 Hz between the ears (top panel) suggests that left- and right-ear sensitivity oscillate in antiphase.

Figure 1 compares the distributions of R^2 obtained from AR(1) simulation (red histogram) with that by shuffling the singles trials (grey histogram). The two distributions almost completely overlap, suggesting that the two methods give equivalent results.

However, when we applied the AR(1) simulation method to the summed sensitivities (bottom panel), where no rhythm is expected (as the sensitivity oscillations in left and right ear should cancel out), the fits of the AR(1) simulated data gave rise to larger R^2 than the shuffled and original data. Note that the fit of the shuffled data remains relatively stable across times series that do contain periodicities (top panel) and those that do not (bottom panel).

Figure 1. Results of fitting the oscillation model to two types of surrogate data obtained by shuffling the original data (black) or by AR(1) simulation (red). In the experiment, participants had to identify the pitch of a lateralised tone embedded in dichotic white noise. The target occurred randomly in the left or right ear. Top panel shows the curve fitting results for the difference in sensitivity (measured by d') between the ears. The bottom panel summarises the results for the sum. The histograms depict the distributions of maximal R^2 , a measure of goodness of fit. The black and red vertical lines indicate the 95 percentiles of the distributions. The blue lines represent the maximal R^2 (at 5.9 Hz for the difference in d') obtained for the original time series. The significant theta oscillation 5.9 Hz between the ears is consistent with the rhythmic attentional sampling hypothesis.

It is also important to point out that failing to find a significant oscillation with this novel method is not the same as proving that no oscillations exist. For that you need model comparisons, preferably with Bayesian statistics such as “Bayes factor” (relative likelihood of there being an oscillation to no oscillation). Alternatively, you could show that the aperiodic temporal structure that is being proposed gives a better fit to the data than does a periodic oscillation, by calculating the ratio of likelihoods of the two models.

Signed

Tam Ho, David Burr, David Alais and Concetta Morrone

Pisa, Toulouse and Sydney, 5/5/2021

Author Rebuttal, first revision:

REVIEWER COMMENTS:

Reviewer #1:

Remarks to the Author:

This is the most important manuscript I have reviewed in several years. It will be highly controversial, and it challenges the dearly held beliefs of many researchers. I can imagine that some researchers will find many things to complain about. For example, they might argue that new methods proposed for assessing oscillations are not optimal (even though they are certainly better than the status quo!). See also Point 4 below.

In my view, however, this paper will have an enormous positive impact on the field, making it possible for researchers to distinguish between periodic and aperiodic temporal structure, both of which are important and have significant implications for the underlying mechanisms of attention. It will not be the last word on this topic — instead, it will lead to many new and important studies. But isn't that what we want for a paper in this journal?

Thank you for the encouraging message! I share your hope that this paper will be one step (but not the last) toward improving our understanding of temporal structure in attention.

1) The issue of whether a pattern of behavioral data is truly oscillatory has also been addressed recently by Yuval-Greenberg and her colleagues (Amit, R., Abeles, D., Bar-Gad, I. et al. Temporal dynamics of saccades explained by a self-paced process. *Sci Rep* 7, 886 (2017). <https://doi.org/10.1038/s41598-017-00881-7>). It would be worth looking at this work to see if there are common solutions. (However, if it turns out to be only tangentially related, the author should not feel obliged to cite it.)

This study by Amit et al. (2017) provides useful context for the current manuscript; it addresses a similar underlying question from a very different methodological perspective. I have added a reference to this study to the manuscript (top third of p. 12)

Along a similar line, it would be worth pointing out (in the Intro or even the Abstract) that the methods developed in this paper could be applied to other cases in which researchers ask whether an underlying process is periodic.

Thank you for this suggestion. I have added a brief note to the introduction (p. 2), and a longer section to the discussion suggesting how this method could be applied to other scientific questions (pp. 13).

2) I may have missed it, but I couldn't find a detailed description of the autoregressive model that was used as one of the alternative analysis methods.

I have added details to the methods section clarifying the autoregressive model, and how it was used to generate the surrogate distribution.

3) Near the top of page 6, it should be mentioned that the new AR method could be used with virtually any correction for multiple comparisons, etc. That is, it is important to note that it has just as much flexibility as the current, flawed approach.

I have added a clarifying note with the initial description of the AR surrogate method (p. 5). I have also added analyses testing the AR surrogate analysis with different corrections for multiple comparisons (p. 9, "Exploring the alternative methods", Fig. 6c).

4) With regard to the simulations shown in Figure 4: I didn't look up these previous studies, but my guess is that they reported oscillations at higher frequencies than were found in the simulations. I can imagine that other reviewers or readers will say that this invalidates the simulations. However, I assume that the particular peak frequency in these simulations will depend on the nature of the aperiodic signal. We don't know the exact nature of the aperiodic signal, so it's not a problem if this particular set of simulations doesn't produce the sample spectrum that was found in the prior research.

The literature often focuses on particular frequencies between 4 and 8 Hz, but it is difficult to compare those frequencies with the peak frequencies of false positives for several reasons.

1. The literature reports behavioral oscillations at a fairly wide range of frequencies (2.5 - 20 Hz). Accordingly, these simulations show false positives at a wide range of frequencies.
2. Studies in this literature often apply preprocessing steps such as smoothing and detrending. These preprocessing steps act as frequency-domain filters, limiting the range of frequencies in which a significant effect is likely to appear.
3. The profile of false positives depends on the characteristics of the aperiodic noise, and these characteristics for behavior are unknown.
4. The file-drawer problem makes it difficult to know whether results that don't match the theory (showing significant oscillations at very low or very high frequencies) would be less likely to be published.

5) On p. 12, where it says "This finding suggests that rhythms in behavior may not be as prevalent as the published literature suggests," I think it would be fairer to say something like "While no analysis can conclusively prove the absence of oscillations, the present results suggest that..."

I agree that this is a fairer assessment of the study. I have updated the text to reflect this change.

6) The discussion of related issues in EEG/ERP research was quite thoughtful, but there are some other methods that might also be relevant in the present context. For a method to determine that multiple cycles are actually present, see:

Watrous, A. J., Fried, I., & Ekstrom, A. D. (2011). Behavioral correlates of human hippocampal delta and theta oscillations during navigation. *Journal of Neurophysiology*, 105(4), 1747–1755. <https://doi.org/10.1152/jn.00921.2010>.

Recent work by Brad Voytek is also relevant, such as:

Donoghue, T., Haller, M., Peterson, E. J., Varma, P., Sebastian, P., Gao, R., Noto, T., Lara, A. H., Wallis, J. D., Knight, R. T., Shestyuk, A., & Voytek, B. (2020). Parameterizing neural power spectra into periodic and aperiodic components. *Nature Neuroscience*, 23(12), 1655–1665. <https://doi.org/10.1038/s41593-020-00744-x>.

However, the author should feel free to disregard these papers if they turn out not to be very relevant or to fit the narrative.

I have clarified the reference to Donoghue et al (2020), and added a reference to Watrous et al (2011) (p. 12).

Reviewer #2:

Remarks to the Author:

I find the results persuasive and therefore accept the paper. However, I suggest to cite a paper

A. Majdandzic et al, "Spontaneous Recovery in Dynamical Networks," Nature Physics 10, 34-38 (2014). Namely, in a rhythmic theory of attention different studies have argued for rhythmic attentional switching between locations at the frequency of the spectral peak. The paper of Majadandzic et al suggested the spontaneous emergence of macroscopic 'phase flipping' phenomena.

Thank you for this suggestion. Unfortunately, I do not understand how this paper ties into the current study. I have used the term "phase" to refer to position within an oscillatory cycle, whereas Majdandzic et al. use "phase" to refer to an overall network state (analogous to the phases of matter). The terms "rhythm", "oscillation", and "periodic" do not appear in that paper. I apologize if I've missed the point!

Reviewer #3:

Remarks to the Author:

I fully agree with the author that the shuffling methodology used presently in the literature cannot be used to provided evidence of rhythmic oscillations (it is unfortunately not the only instance in the literature where the shuffling or permutation schemes do not correspond to the intended purpose). And the fact that the 11 significant results from the literature (out of 23 tests) became

all non significant when compared to an (arhythmic) AR(1)-based null distribution is quite convincing that the simplistic shuffling procedure possibly misled important scientific findings.

Thank you for the careful reading of this manuscript, and for the thoughtful suggestions for how to improve it.

Although the proposed solutions are clearly better than the actual practice, I'm a little less enthusiastic concerning the two proposed solutions for the following reasons:

- AR(1) is just one possible aperiodic signal out of so many possible alternatives

I agree that the AR(1) is only one of many types of aperiodic structure that could be tested. From the original manuscript: "Both of these methods [the AR(1) and robust est. analyses] could also be adapted to examine 'power-law' structure in behavior by fitting the data to a $1/f^\beta$ spectrum instead of to an AR(1) process." I have added a section to the Discussion clarifying how this method could be adapted to account for other types of aperiodic structure, such as $1/f$ noise (p.12-13).

I have also clarified the justification for using AR(1) processes. This study focuses on AR(1) models because they offer a simple, clear null hypothesis against which we can test for oscillations in behavior. The AR(1) model is not intended to fully capture the dynamics of behavior. Instead, the AR(1) model provides the null hypothesis we are trying to rule out. This replaces the current null hypothesis from shuffling-in-time, which holds that the data show no temporal structure whatsoever. I have clarified these points in the manuscript (p. 5, Results, "Distinguishing between periodic and aperiodic temporal structure"; p. 11).

- AR(1) is stationary and the author acknowledges that attentional switching data are probably often non-stationary.

Thank you for bringing this point to my attention. I had used the term "non-stationary" in a non-technical sense to mean "not sustained uniformly over time." I have fixed this issue in the manuscript.

Note that an AR(1) process is stationary over an infinite number of samples, but a single given time-series generated by an AR(1) process often appears non-stationary, with a mean that changes over time. This is especially true of short time-series (as we have in studies of behavioral oscillations).

- AR(1) results are probably very sensitive to the chosen sampling rate. Doesn't AR(1) behave quite differently if one uses only half of the time-points or on the contrary twice as much? Other approaches based on fitting the spectrum (as mentioned p15) or equivalently on the ACF (see e.g. Abrahamsen) might be more robust to sampling rate choices?

This is an interesting point. To address these questions, I have performed new simulations examining the effect of sampling rate on the rates of Type I and Type II error for the AR surrogate, robust est., and LF2012 analyses. These new analyses have been added to the results section of the manuscript (p. 9, "Exploring the alternative methods", Fig. 6g-i).

As you predict, the sampling rate influences the sensitivity and rate of false positives for both the alternative analyses -- even though the robust est. analysis is based on fitting the spectrum. However, for all sampling rates tested (which span the range of experimentally relevant frequencies), both of the alternative methods give a lower rate of false positives than shuffling in time. The AR surrogate method is especially well suited to lower sampling rates. The robust est. method, in contrast, is the only method that controls the false positive rate at higher sampling rates.

These findings suggest that, when designing a new study of behavioral oscillations (before collecting any data), researchers should use these simulation tools to select the best experimental paradigm and analysis method for their study. This will ensure that future studies control the rate of false positives while maximizing experimental power.

I encourage the author to propose a more global approach, e.g. by first evaluating (visually or by a test) whether the stationarity assumption is tenable (e.g. by looking at a time-frequency plot like the continuous wavelet transform), second to check with a few more complex models than an AR (1) and third to discuss more on the fact and consequences that other aperiodic models could still modify the results. The above suggestions are just ideas and more sophisticated methods based on the extremely vast literature on time-series might be even more adequate.

I'd like to first clarify the goals and conclusions of the current study. The AR(1) model is not intended to fully capture the dynamics of attention. Instead, the AR(1) model is proposed as a null hypothesis to test against the data. This replaces the null hypothesis that is currently used (and usually implicit) in the literature: that the data have no temporal structure whatsoever. I have clarified this point in the manuscript (p. 5; p. 11), and I have outlined how these methods could be adapted to test for oscillations against other types of aperiodic structure (p. 12).

It is critical that the field works towards a more nuanced understanding of the temporal structure of attention and perception. An approach like you suggest will help to develop this understanding. For the questions addressed by the current manuscript, however, the simplicity of the AR(1) model is a virtue, not a fault: it allows us to test a simple null hypothesis, and to find that published data-sets do not allow us to reject that null hypothesis.

Technical issues:

- In the novel approach, a cluster-based permutation test is proposed to tackle the multiple comparison problem. I see however two difficulties here. First, as stated e.g. in the reference

[34], when a cluster is significant, one cannot state that each point within the cluster is significant (but only that there is a point within the cluster that is significant). So it would be incorrect in the interpretation and in the simulation to act as if the frequencies within a significant cluster are all significant.

This is absolutely correct; this point is important and commonly misunderstood in cluster permutation tests. I hadn't realized until thinking carefully about this comment that I made the same mistake! Thank you for bringing this to my attention.

I implicitly made this mistake when I examined the error in the frequency of identified oscillations (formerly Figs. 4 & 6, now Fig. 4). The original analysis counted as significant all points within each cluster that exceeded the significance threshold. I have changed this analysis to more accurately reflect the inferences licensed by a significant cluster test. The cluster-permutation procedure tests for exchangeability between the two conditions. In the AR surrogate method, a significant cluster test tells you that the behavioral spectrum is not compatible with the spectra generated by an AR(1) process. To find the important frequencies, you can then visually examine the spectrum to identify peaks. This follows the recommended use of cluster-based tests in electrophysiology:

https://www.fieldtriptoolbox.org/faq/how_not_to_interpret_results_from_a_cluster_based_permutation_test/

In the updated analyses, instead of taking every significant sample, I have computed the peak frequency from each significant experiment (an experiment is "significant" when it has a minimum corrected p-value of less than .05). This procedure allows us to compare across methods that use different approaches to correct for multiple comparisons.

The original analysis showed low accuracy for shuffling-in-time because those methods yield statistically significant samples at a large number of frequencies; in this updated analysis, the estimated peak frequencies are highly precise and accurate (within 0.3 Hz) for every analysis method. The mean error in the recovered frequency is smaller than the frequency resolution of the AR surrogate and robust est. analyses. I have added a section to the results, updated the figures (Figs. 3 & 4) to show the peak frequency instead of all frequencies within a significant cluster. I have removed the table comparing accuracy between analysis methods, because all the analysis methods are highly accurate when we focus on peak frequency instead of including all "significant" samples.

Second, the assumptions for the construction of the null distribution of a cluster-based permutation rely on invariance of correlation between equidistant points, i.e. some sort of stationarity. Here the

points are frequencies and I do not see reason to believe in stationarity across frequencies of spectral estimations.

I cannot find any reference for the idea that cluster-based permutation tests require invariance of correlation between equidistant points. The use of cluster tests in this study mirrors thousands of studies using cluster tests to investigate time-frequency

representations (TFRs) or event-related potentials (ERPs), both of which lack invariance of correlation between equidistant points. For example, TFRs often show a strong oscillation at a single frequency band, leading to a lack of invariance between equidistant points (correlations are strong within the frequency band, but weak outside it). As a second example, ERPs often rise or fall over time, showing a lack of stationarity in the changing mean of the signal. Cluster-based permutation tests are the

state-of-the-art method for analyzing data with arbitrary spectral and temporal characteristics.

For completeness, I have simulated the results when using different methods of correcting for multiple comparisons (p. 9-10, Fig 6b-c). The rate of Type I errors is only adequately controlled when using cluster tests.

- to allow us to better understand if the difference in the results are due to the core procedure or to the subsequent multiple comparison methods, the simulations should as much as possible separate the two: first compare the results when no multiple comparison corrections are applied, second, when possible, use the same multiple comparison corrections and finally compute the results of the full methods.

This suggestion substantially improves the manuscript. To help understand how shuffling in time leads to false positives, I have added plots of the uncorrected significance for each frequency bin (Fig. 3b-e). When data are analyzed using shuffling-in-time, the false positive rate of single frequency components substantially exceeds .05. I then show histograms of the

peak frequencies identified in experiments with a significant false positive result after correcting for multiple comparisons across frequencies (Fig. 3f-i).

I have also included a section describing the results when the AR surrogate and robust est. analyses are performed using other methods of correcting for multiple comparisons (p 9-10, Fig. 6b-c). I have added these plots after the main results, to maintain focus on the main idea of the paper: that shuffling in time cannot distinguish between oscillatory and aperiodic temporal structure. For the AR surrogate analysis, only cluster-based permutation tests adequately control the rate of false positives. For the robust est. analysis, the choice of correction method does not strongly influence the results.

- the method and the exact simulation settings should be clearer: for example, (1) to compute the rate of false positives do you consider a sample significant as soon as one frequency is significant or is it the average of significance over all frequencies/tests ? (2) if a cluster is significant, how do you treat each point within the cluster ?

I have clarified these points in the manuscript. To summarize:

(1) An experiment is counted as a positive result if it shows any significant peaks in the spectrum after correcting for multiple comparisons across frequencies -- in other words, when the experiment has at least one significant sample after correcting for multiple comparisons. This procedure is consistent with the way

studies are analyzed and reported in this literature. This information has been added to the Methods section (p. 14).

(2) I have updated these analyses to select the *peak frequency* (instead of *every significant sample*).

(3) I have clarified the methods of the analysis methods and simulations.

- I am surprised by the fact that as stated in bottom of page 18, the author used k=500 shufflings. It implies that the smaller raw p-value is $1/500=0.002$. If I understand correctly the settings, a Bonferroni

correction on 256 tests is then applied, which means that the smaller possible corrected p-value is $0.002 * 256 = 0.512$ (!), so that no results can be significant ! So first I'm surprised to see significant results in these simulations and second, the number of shufflings should be increased so that the smallest possible corrected p-value is at the very very most 0.025.

I completely agree -- I was also surprised when I realized this about the method used by Landau and Fries (2012). The issue arises in the way the authors defined their p-values when the empirical value is larger than *every shuffled value*. In those cases, as you suggest, many authors set $p = 1/k$. Other authors, however, simply define the p-value as $p = \text{mean}(\text{permuted} > \text{empirical})$. Under this definition, $p = 0$ when the empirical value is larger than every permutation. And of course, a Bonferroni correction has no effect when $p = 0$.

LF2012 used $k=500$ permutations, along with a Bonferroni correction for multiple comparisons across frequencies. Furthermore, they do not report exact p-values for these analyses, but only report whether $p < .05$. As you point out, this means they must have set $p = 0$ when the empirical value was stronger than every permutation. The analyses I simulate here replicate LF2012; I have preserved their procedure to contrast it with other methods. As the simulations demonstrate (Fig. 2, Fig. 4b-e, Fig. 6b-c), the crucial factor leading to false positives is shuffling in time, not this idiosyncrasy of Bonferroni correction. I have clarified this point in the methods section (p. 15).

Although I share your hesitancy about this procedure, there is some precedent for it in the field. For example, in the article popularizing cluster-based permutation tests, Maris and Oostenveld (2007) describe one result like this: "In fact, its Monte Carlo p-value was zero; none of the 1000 random partitions resulted in a cluster-level statistic that is larger in absolute value" (Section 3.1.1, 2nd-to-last paragraph). They do not suggest correcting this p-value to $p = 1/k$. This is not to argue in favor of the analysis procedure used by LF2012, but to note that the field lacks clarity about the best practice in this case.

- In the simulation results of Fig4a, why are the error rates of AR-based methods close to 0 instead of being close to 5% ? Does it imply they are highly conservative approaches?

The AR surrogate and robust est. methods show very low rates of false positives when data are generated as aperiodic noise. These approaches are conservative relative to an

(unknown) optimal analysis method. But importantly, the alternative methods must be preferred over existing methods, which have drastically inflated rates of false positives.

I have added a frequency-limit to the AR surrogate analysis (Fig. 6a); this limit reduces the conservatism of this method, and increases its sensitivity (Fig. 2a, 6a). I have clarified the issue that these methods are conservative relative to an unknown optimal technique (p. 7).

- I did not find which multiple correction procedure is proposed (and used in the simulations) for the second proposal.

Thank you for pointing out this oversight. In the robust est. method, I have corrected for multiple comparisons using Bonferroni corrections. This point has been added to the methods section.

I have also added simulations showing how these results depend on the method of correcting for multiple comparisons (p. 9, Fig. 6b). Results are similar whether p-values are corrected using Bonferroni corrections or FDR.

more minor comments:

p5 “This AR(1) model captures the aperiodic structure – but not the periodic structure – of the time-series”. This claim is too strong.

Thank you for drawing my attention to this. As you suggest above, behavior may show other types of temporal structure that are not captured by an AR(1) process (like the time-dependence of errors in an ARMA(p,q) process). I have amended this sentence to read: “The AR(1) model captures the lag-1 autocorrelated aperiodic structure -- but not the periodic structure -- of the time-series.”

p15 “The alternative analyses proposed here test the null hypothesis that the data are autocorrelated” is slightly an over statement since it does not cover AR(p)

That is correct -- thank you for this close reading. I have updated this sentence to read: “The alternative analyses proposed here test the null hypothesis that the data are generated by an AR(1) process.”

p15 “However, the simulations and analyses reported here are consistent with the now-widespread conclusion that attention is non-stationary “: I don’t see how this can be true since the AR(1) is stationary.

It is more accurate to say that attention is “not sustained uniformly over time”, rather than “non-stationary”. I have changed the text to reflect this point.

Note that an AR(1) process is stationary if seen over a large number of time-points, but that brief time-series generated by an AR(1) process will often appear to be non-stationary (with a mean that changes over time).

p14 I’m not sure what is meant by “lower-frequency aperiodic activity”

I have clarified this sentence to read: “A similar pattern emerges in non-oscillatory scale-free activity: the power of higher-frequency neural activity depends on the ‘phase’ of aperiodic lower-frequency activity (He et al. 2010).”

p17: “All p-values reflect two-tailed tests”: does it mean that frequency values that are lower than the null distribution are deemed significant ?

The original phrasing was unclear: "All p-values reflect two-tailed tests except where otherwise noted." This sentence was intended to refer to comparisons between analysis methods. Within each analysis method, the tests searched for *peaks* in the spectrum above the surrogate distributions (using one-tailed tests). I have removed this sentence and specified whether each test was one- or two-tailed.

Reviewer #4:

Remarks to the Author:

This is an important article, relevant to a large audience. The article is well written and easy to understand. The conclusions drawn are well motivated and supported by the simulated data and re-analyses of published datasets.

Major

While I appreciate the logical structure of the article, there is one obvious question that remains somewhat unaddressed. It is based on the fact that the data simulations test false positive and true positive rates for the analysis approaches of two highly influential studies in the field and compare them to the alternative approaches. The final results section, however, reveals false positives in actual data of four other studies (which are arguably less influential and present generally somewhat weaker evidence for the temporal modulation of attentional sampling). After reading the article, the reader is thus left alone with guessing/wondering whether the results of the two highly influential articles (LF2012, FSK 2013) present false positives or veridical periodic modulation of attentional sampling. In case the authors of LF2012 and FSK 2013 do not agree with sharing their data for a re-analysis in the present article, this should be stated in a very transparent way.

I requested both of these datasets from the authors. Ian Fiebelkorn did not provide data for FSK2013. Ayelet Landau provided data for LF2012, but the two of us could not

reconstruct the exact time-series plotted in the original paper. I have added a note about this to the methods section (p. 18).

A major problem of published articles on rhythmic attentional sampling is also the fact that the rhythmic modulation is typically observed in a very limited time interval (usually up to 1s). The author might want to demonstrate to what extent (if any) the data simulation results depend on the length of the sampled time window. This would be very helpful for the audience, especially with regard to the design of future experiments.

Thank you -- this suggestion improves the manuscript by illustrating how a different experimental design influences our ability to detect a real oscillation. I have added simulations of experiments in which the time window is halved or doubled (p. 9, Fig. 6d-f). For brief signals (and signals of the typical length in this literature), the AR surrogate method has a higher sensitivity than the robust est. method. For long signals, however, the AR surrogate method does not adequately control the rate of false positives.

These findings suggest that, when designing a new study of behavioral oscillations (before collecting any data), researchers should use these simulation tools to select the best analysis method for their study. This will ensure that researchers can control the rate of false positives while maximizing experimental power.

One approach that is sometimes considered an alternative to 'shuffling in time' is 'shifting in time'. Shifting in time does not remove the temporal structure in the data but instead it abolishes any phase-locked effects. While 'shifting in time' is certainly not always a sensible analysis approach, it can be used in case the phase-locking (and/or periodic relation) of two time-domain signals (e.g., neural and behavioral sampling of a stimulus) is tested. It would be helpful if the author could refer to this approach as well.

I have added a section discussing 'shifting in time' and related statistical techniques (Supplementary information, p. 4).

I was not entirely sure but it seems the analysis code is available in python. In order to make it accessible to a broader audience, it would be great if it would be available for matlab as well.

In recent versions of Matlab, Python functions can be called directly from within Matlab. https://uk.mathworks.com/help/matlab/matlab_external/call-python-from-matlab.html

Signed comments from Davidson, Tsuchiya, van Boxtel

We thank the editors for this opportunity to provide a signed comment. In this paper, a widely-used permutation method which shuffles the temporal order of events to create a null-distribution of time-series data is challenged. The paper deduces that false positives

may be high using this method, because other methods (discussed by the author) do not find true positives.

This is not an accurate summary of the reasoning in the manuscript. The paper *does* conclude that shuffling in time results in false positives, but this conclusion *does not* rely on simulations with other methods; it is based on simulations using shuffling in time. Furthermore, the AR surrogate and robust est. methods do identify true positives, and they do so with much higher selectivity than the standard methods of shuffling in time. The sensitivity of any of the analysis methods is independent from the finding that shuffling in time leads to drastically inflated rates of false positives.

While we think this is an important paper that will help move the field forward, we have several concerns regarding the interpretation of, specifically, our previously published dataset, and, generally, the sensitivity of the alternate methods proposed.

Thank you for the thoughtful reading! It is true that the alternative analysis methods do not catch every true effect -- however, this is true of every statistical test in existence. For example, a t-test will not reliably identify a weak effect with a small sample size. The AR surrogate and robust est. analyses are the only analyses that control the rate of Type I errors, so they must be preferred over shuffling in time. I have updated the AR surrogate method by designating an *a priori* frequency range of up to 15 Hz. This increases the sensitivity of the AR surrogate method (Fig. 6a). To maximize experimental power and minimize the risk of false positives, future studies can increase the length of the behavioral time-series or the behavioral sampling rate, and use the robust est. analysis (see Fig. 6d-i).

Our data are tested against these two methods, after first subjecting the raw time-series to the pre-processing pipeline of other papers (LF2012 or FSK2013) which includes mean subtraction and linear detrending, steps we did not originally perform.

This is not accurate. The re-analyses did not reproduce the pre-processing steps of LSF2012 or FSK2013 (which differ from each other). That being said, both the AR surrogate and robust est. analyses do detrend the signals before computing the DFT. Detrending helps improve the interpretability of the Fourier coefficients, and is standard practice in digital signal processing.

From the provided code, it appears the empirical peaks are still above the (uncorrected) 95% CI of the AR surrogate analysis. However, these peaks are singular, and thus do not survive the 1-dimensional cluster correction for adjacent significant peaks in the spectrum (Mismatch; $f = 3.37$ Hz, $p = .0097$; Match; $f = 8.08$ Hz, $p = .0037$). The Robust est. method does not return significant peaks.

When an analysis involves multiple statistical tests, it is essential to correct for multiple comparisons. With 22 frequency bins, we would expect to find 1 significant sample in each experiment by chance. These p-values are only interpretable after correcting for multiple comparisons.

It's important to note that our method/pipeline is different to LF2012 or FSK2013 , and was carefully selected due to the highly unique nature of our dataset -- unique even in the context of attentional sampling research. We do shuffle in time, but do not detrend, Hanning taper, or otherwise smooth our analysis as is usually performed. We deliberately computed the fft with a single taper to maximise frequency-resolution [...]

As illustrated in Fig 2, the critical step in introducing false positives is shuffling in time. Preprocessing can constrain the range of peak frequencies, but it is not the crucial factor in leading to false positives.

[...] and included important negative control conditions to test for the absence (or change) in attentional sampling frequency consistent with theory.

More specifically, we report a difference in frequency between conditions (mismatch and match), and absence in strong control conditions (visual only - no cues, attended vs unattended), as well as convergent EEG data.

A control condition does not change the fact that shuffling in time inflates the rates of false positives. Note also that Davidson et al. (2018) did not statistically test for a difference between the two conditions -- the significant peak in one condition and the lack of a significant peak in another condition. As a consequence, it is not valid to interpret differences between these conditions (Gelman & Stern, 2006).

Importantly, we also do not investigate accuracy over time by systematically varying the SOA between a target and probe, as is standard in attentional sampling research.

Although the theoretical framing of my study focuses on accuracy and attentional switching, the statistical conclusions hold for any attempt to discriminate oscillations from autocorrelated noise in any time-series. These methods are agnostic as to how the time-series is obtained, or what it represents.

As a result, we do not have an equivalent amount of data at each time-point, and by design, we have a large aperiodic signal in our time-series, as the proportion of first-switches has a heavy positive skew. This clustering of data reduces the estimated likelihood of detecting a sustained oscillation at later time-points, and reduces the sensitivity of our data when subjected to the Robust est. method (more below).

There is no reason to believe that shuffling-in-time is a better choice for this experiment than the AR surrogate or robust est. analyses. If these alternative analyses cannot detect oscillations in this dataset, then the experiment must be re-designed and performed with higher experimental power. Simulations like those in this manuscript -- performed prior to collecting any data -- will help ensure that future experiments are capable of revealing oscillations in the data they propose to collect. I have clarified these points in the manuscript (Fig. 6; p. 11).

Without supplying the highly unique context of our original design, the author misrepresents the likelihood of the new alternate methods of detecting an oscillation when reanalysing our dataset. This point is clear when investigating the simulations provided in the manuscript (Table 1 and Figure 6). Although the author states that both methods “successfully recover true oscillations in simulated behaviour”, this is only reliably the case at higher frequencies (above 4 Hz), and when the amplitude of oscillations is above 0.3. As we focus on low frequencies (below 4 Hz) and have small oscillations (amplitude < 0.3), these new methods would be unable to recover true oscillations, even in the simulated data.

I do not believe that I have misrepresented the sensitivity of the alternative analyses. Just as a t-test cannot reliably identify very small effects in large amounts of noise, the AR surrogate and robust est. methods cannot identify weak oscillations in a brief, noisy time-series. We could frame this issue as a problem with the statistical tests, or as a problem with the experimental design. In either case, it remains true that previous studies have not provided evidence for behavioral oscillations in attentional switching.

I have updated the comparisons between methods to consider all frequencies and amplitudes, not only frequencies $\geq 3\text{Hz}$ and amplitudes ≥ 0.2 .

It is also important to note that the Robust est. method may not be as applicable to oscillations in behaviour - owing to the assumptions of that method that were determined for application in climate science (Figure 2). In particular, the multi-taper method of Mann and Lees (1996) invokes assumptions “that are faithful to our understanding of the physics governing the climate system”, such as isolating peaks only with a coherent phase spectrum, and identifying relatively large amplitude oscillations. It is entirely plausible that attentional sampling is quasi-rhythmic, rather than being determined by a clock-like or climatic oscillation, and that the depth of this oscillation tapers with attentional focus. This interpretation of a short lived burst of attentional sampling is consistent with our (and others) data, but would not be detected by the Robust est. method.

It is important to keep in mind that the robust est. analysis was designed for one specific feature of climate data: the autocorrelated noise background. Behavioral data are also autocorrelated, making the robust est. method equally appropriate for behavioral and climate data. Davidson and colleagues raise 2 specific objections to the robust est. method.

1. Davidson and colleagues claim that the robust est. analysis is designed for strictly rhythmic oscillations, whereas behavioral oscillations may be quasi-rhythmic. This is not accurate; the robust est. analysis can also detect quasi-rhythmic oscillations. Mann & Lees (1996) specifically discuss several different quasi-periodic rhythms in different climatic datasets, and state: “Our

red-noise significance criterion, however, isolates the same peaks as significant whether they are considered periodic or quasiperiodic” (p. 428).

2. Davidson and colleagues claim that the robust est. analysis is restricted to high-amplitude oscillations. This is akin to claiming that a t-test is restricted to large-amplitude effects. The robust est. analysis can detect any arbitrarily small oscillation, as long as it is reliably higher than the analytic AR(1) spectrum.

The overly-conservative nature of these tests is also evident in Table 1 in the paper, which displays the proportion of false-positives for each analysis method. A well calibrated method would produce false positives at the expected rate of 0.05. However, both the Robust and AR surrogate methods are appreciably below this cut-off. For unstated reasons, the author only performed one-tailed tests, showing significantly “greater” than .05 in bold. We believe it is only fair to perform two-tailed tests and report their results, revealing the conservative nature of these results. The author is then expected to comment on this property of the alternative methods.

I performed 1-tailed tests because the most important thing about a statistical analysis is that it controls the Type I error rate. A conservative test is not optimal, but it must always be preferred over a test that leads to false positives. I have clarified this point in the manuscript (p. 7; p. 17).

Furthermore, the AR surrogate tests have been updated to only analyze frequencies up to 15 Hz. This test is no longer conservative when it comes to identifying oscillations in the presence of random walk noise.

In summary, while we agree that clarification and improved methods are needed to establish true ‘oscillations’ in behaviour, the new proposed methods, however, appear overly conservative for the depth of oscillations typically reported in our field. Additionally, our study is particularly unique, with features that decrease the likelihood of detecting oscillations with these new methods - as is evident in the authors’ own simulations.

I see no reason to believe that these data are poorly-suited to analysis with the AR surrogate and robust est. methods. If these methods cannot detect any oscillations in

the data, then no currently existing method can do so while controlling the rate of false positives. Regardless, it remains true that no existing analysis can uncover evidence for behavioral oscillations in this dataset.

These caveats are critical to restate when introducing the reanalysis of our (and other authors' data). The author should state what the expected likelihood of detecting 'true' oscillations is in the reanalysis of datasets - based on the frequency range and depth of oscillations we have reported. When provided with these caveats, it is unsurprising that with a different fixed preprocessing pipeline, and more conservative methods, oscillations are not recovered.

Because of the severely inflated Type I error rate, a positive result from

shuffling-in-time cannot be interpreted as evidence for oscillations in behavior. I have shown that we find no evidence for behavioral oscillations in this dataset using any existing analysis method that controls the rate of Type I errors. By analogy, let's say we want to test whether the mean value of a normally-distributed variable is different from 0. If we expect an effect size of 0.00001, but only collect 10 observations, we will be unlikely to detect an effect in this experiment. In this case, the only way to boost experimental power is to design a new experiment. I have clarified in the document (p. 9; p. 11) that these results do not comprise evidence for a lack of oscillations -- instead, we have a lack of evidence for oscillations.

Signed comments from Ho, Burr, Alais, and Morrone

The autoregressive modelling method proposed by the author is typically applied to historical data, such as in climate science, where accurate predictions about the future rely on past measurements. Here it is being applied in a relatively novel way to our binned data, which have been sorted and

grouped relative to a reset signal, rather than in chronological order. This is not necessarily wrong, but interpretations are less immediate.

The important feature of the data is that they are autocorrelated over time. The AR surrogate analysis is agnostic about the units of time -- it could be millenia from the present day, or milliseconds from a cue stimulus (both of which represent a chronological order).

We applied his proposed method to our data and found that all five oscillations reported in Ho et al. (2017) had a goodness of fit (R^2) larger than 95% of the AR(1) simulations at the reported theta and alpha frequencies. The only condition where we obtained a non-significant result when compared against AR(1) noise was also non-significant in the original publication.

First, we should note that Ho and colleagues did not apply either of the methods I proposed. Instead, it appears they propose the following novel analysis technique:

- Fit the behavioral time-series to an AR(1) process
- Generate a surrogate distribution using the fitted AR(1) model
- For each empirical and surrogate time-series
 - Find the R^2 between the data and the best-fitting sine-wave
- Find the p-value by comparing the empirical R^2 against the surrogate R^2 distribution

This is an intriguing proposal. It would be important to know if this method shows improved sensitivity over the AR surrogate and robust est. methods. When I perform the procedure outlined above, however, I do not find a significant result in their data. I would encourage Ho and colleagues to develop this method and examine how it performs relative to the AR surrogate and robust est. methods. In order to interpret the results of this test, it will be necessary to ensure that this method controls the rate of false positives.

Furthermore, it would be necessary to clarify how the null hypothesis of this test differs from DFT-based tests. DFT-based approaches (all of the methods I have examined) test whether

the data includes a higher spectral peak than would be expected by chance (in an AR(1) process or in data with no temporal structure, depending on the analysis method). The R^2 approach, by contrast, tests whether the data can be approximated by a sine wave more closely than would be expected by chance.

It is also important to point out that failing to find a significant oscillation with this novel method is not the same as proving that no oscillations exist.

I agree that this paper does not demonstrate that no oscillations exist in attention. I have updated the manuscript to clarify this point. This section of the results now reads: "Although no statistical analysis can conclusively prove the absence of oscillations, the present results suggest that putative rhythms in behavior could be explained by aperiodic temporal structure." (p. 9)

For that you need model comparisons, preferably with Bayesian statistics such as "Bayes factor" (relative likelihood of there being an oscillation to no oscillation). Alternatively, you could show that the aperiodic temporal structure that is being proposed gives a better fit to the data than does a periodic oscillation, by calculating the ratio of likelihoods of the two models.

Bayesian statistics do not solve this problem. Bayesian support for the null hypothesis would let us conclude that there are no oscillations in one particular study, but it would not let us generalize to behavioral oscillations in general.

References

Gelman, A., & Stern, H. (2006). The difference between "significant" and "not significant" is not itself statistically significant. *The American Statistician*, 60(4), 328-331.

Author Rebuttal, first revision:

Decision Letter, second revision:

22nd October 2021

Dear Dr. Brookshire,

Thank you for submitting your revised manuscript "Putative rhythms in attentional switching can be explained by aperiodic temporal structure" (NATHUMBEHAV-210314746B). It has now been seen by the original referees and their comments are below. As you can see, the reviewers find that the paper has improved in revision.

I am therefore delighted to say that we will be happy in principle to publish it in Nature Human Behaviour, pending minor revisions to satisfy the referees' final requests and to comply with our editorial and formatting guidelines.

We are now performing detailed checks on your paper and will send you a checklist detailing our editorial and formatting requirements within two weeks. Please do not upload the final materials and make any revisions until you receive this additional information from us.

Sincerely,
Jamie

Dr Jamie Horder
Senior Editor
Nature Human Behaviour

Reviewer #1 (Remarks to the Author):

This is an excellent revision that satisfactorily addressed the issues raised in my original review. This paper will have a substantial impact on an important area of research, and I am enthusiastic about it being published in this journal.

Steve Luck (signed review)

Reviewer #2 (Remarks to the Author):

The author replies in agreement with my recommendations and reading the suggestions of other referees I warmly suggest to accept the manuscript

Reviewer #3 (Remarks to the Author):

(I was referee no 3) The author modified the manuscript and added simulations for most of my comments and replied convincingly to the others. I believe that the manuscript is now very interesting and comprehensive and I recommend it should be accepted. I have two very minor comments:

- "[AR1] discriminate between periodic and aperiodic structure in behavioral time-series" (or similar sentences: abstract, 1st paragraph of intro, last paragraph of intro, first paragraph of "Distinguishing between periodic ...") seem to strong since AR1 does not cover other aperiodic model. I would also recommend that one sentence about this limitation of interpretation/ discussion about the aim of the proposed models is given in the abstract

- (p5) "The novel first method is based on randomization" is formally incorrect, since AR surrogate are generated by simulated (pseudo) random noise and not shuffled/permutated noise. In the statistic literature it would be called "a parametric bootstrap method".

As a note to the author, and not as a comment that would ask to modify the manuscript: even if it not explicitly written in the literature, it is clear that the cluster mass test do have assumptions on the signals in order to provide a genuine distribution that is supposed to be usable for clusters wherever they are on the interval. And I believe the minimal assumptions are the one I mentioned. The degree to which departing from these assumptions (as its use in the present manuscript I believe) alters the false positive rate would be an interesting topic of research. It does certainly not weaken your proposed approach.

Olivier Renaud (signed review)

Reviewer #4 (Remarks to the Author):

The author addressed my own previous concerns. Also, I think that Figure 6 in the revised manuscript is indeed helpful for planning future studies.

In the meantime, I had the chance to try out some analyses with the AR(1)-approach myself and I must confess that I now share some of the concerns mentioned by Reviewer #3: "Although the proposed solutions are clearly better than the actual practice, I'm a little less enthusiastic concerning the two proposed solutions for the following reasons: AR(1) is just one possible aperiodic signal out of so many possible alternatives."

I totally see that the article does not claim that the alternative analysis approaches pose final solutions to the problem. However, as we can also see in Figure 6, these approaches have obvious

disadvantages as well. For, for a typical experiment with a relatively short time interval analysed for attentional switching (< 1s) with a limited sampling rate (< 30 Hz), the alternative approaches do not provide satisfactory true positive rates.

At present, I think that individual parts of the article differ regarding their potential impact. First, the article makes a very relevant contribution because it reviews and nicely illustrates an important and pressing problem. It is timely to raise awareness for this in order to prevent invalid interpretation. Second, it proposes alternative analysis approaches (which are probably the most well-suited available to our knowledge) but these come with some disadvantages, too. Third, the re-analysis of previous studies using the alternative approaches is insightful but not entirely conclusive given the disadvantages of the alternative approaches and the fact that the arguably most clear-cut examples of rhythmic attention switches in the literature (e.g., Fielbelkorn et al 2013, *Current Biology*; Helfrich et al 2018, *Neuron*) are not among the re-analysed ones.

Final Decision Letter:

Dear Dr Brookshire,

We are pleased to inform you that your Article "Putative rhythms in attentional switching can be explained by aperiodic temporal structure", has now been accepted for publication in *Nature Human Behaviour*.

Please note that *Nature Human Behaviour* is a Transformative Journal (TJ). Authors whose manuscript was submitted on or after January 1st, 2021, may publish their research with us through the traditional subscription access route or make their paper immediately open access through payment of an article-processing charge (APC). Authors will not be required to make a final decision about access to their article until it has been accepted. IMPORTANT NOTE: Articles submitted before January 1st, 2021, are not eligible for Open Access publication. Find out more about Transformative Journals

With best regards,

Jamie

Dr Jamie Horder
Senior Editor
Nature Human Behaviour